# GENERATIVE UNIVERSAL VERIFIER AS MULTIMODAL META-REASONER

**Xinchen Zhang**[1,2]   **Xiaoying Zhang**[2]   **Youbin Wu**[2†]   **Yanbin Cao**[2]   **Renrui Zhang**[2]
**Ruihang Chu**[1]   **Ling Yang**[3]   **Yujiu Yang**[1†]   **Guang Shi**[2]
[1]Tsinghua University   [2]ByteDance Seed   [3]Princeton University
zhangxc24@mails.tsinghua.edu.cn
youbinwu@bytedance.com, yang.yujiu@sz.tsinghua.edu.cn
https://omniverifier.github.io/

## ABSTRACT

We introduce *Generative Universal Verifier*, a novel concept and plugin designed for next-generation multimodal reasoning in vision-language models and unified multimodal models, providing the fundamental capability of reflection and refinement on visual outcomes during the reasoning and generation process. This work makes three main contributions: (1) We build **ViVerBench**, a comprehensive benchmark spanning 16 categories of critical tasks for evaluating visual outcomes in multimodal reasoning. Results show that existing VLMs consistently underperform across these tasks, underscoring a substantial gap from human-level capability in reliable visual verification. (2) We design two automated pipelines to construct large-scale visual verification data and train **OmniVerifier-7B**, the first omni-capable generative verifier trained for universal visual verification and achieves notable gains on ViVerBench(+8.3). Through training, we identify three atomic capabilities in visual verification and demonstrate how they generalize and interact synergistically. (3) We propose **OmniVerifier-TTS**, a sequential test-time scaling paradigm that leverages the universal verifier to bridge image generation and editing within unified models, enhancing the upper bound of generative ability through iterative fine-grained optimization. Beyond generation, we extend universal verifier to broader world-modeling interleaved reasoning scenarios. Empirically, OmniVerifier-TTS achieves improvements on T2I-ReasonBench(+3.7), and GenEval++(+4.3), outperforming existing parallel test-time scaling methods, such as Best-of-N. By endowing multimodal reasoning with reliable visual verification, OmniVerifier advances both reliable reflection during generation and scalable test-time refinement, marking a step toward more trustworthy and controllable next-generation reasoning systems.

## 1 INTRODUCTION

The field of multimodal large language models (MLLMs) is undergoing a remarkable revolution in its application scenarios, evolving from simple image understanding (Liu et al., 2023a; Li et al., 2023; Zhu et al., 2023; Lu et al., 2024; Zhang et al., 2024a) to complex visual reasoning tasks (Guo et al., 2025; Comanici et al., 2025; OpenAI, 2025; Zheng et al., 2025; Li et al., 2024). Vision-language models (VLMs) (Guo et al., 2025; Team et al., 2025a; Bai et al., 2025; Team et al., 2025b; Seed, 2025) and unified multimodal models (UMMs) (Deng et al., 2025; Xie et al., 2025b; Liao et al., 2025; Wu et al., 2025b; Yang et al., 2025a) integrate vision and language, enabling more comprehensive cross-modal interactions and paving the way toward a more intelligent reasoning and generation system.

Toward next-generation multimodal reasoning, we argue that self- and externally-critique will be key drivers of model advancement. In interleaved scenarios, effective progress requires verifying not only textual answers but also visual outcomes, which refer to any model-generated visual artifact, such as images in text-to-image generation or intermediate visual states produced during

---

†Corresponding authors.

stepwise tool-call reasoning. Unlike textual outputs, these visual outcomes are high-dimensional and ambiguous, making their reliable assessment difficult without explicit verification. We therefore contend that visual-outcome verification is fundamental to scaling multimodal reasoning and generation, enabling MLLMs not only to generate outcomes but also to iteratively understand, verify (Xiong et al., 2025; Wang et al., 2025b), and refine (Madaan et al., 2023; Kumar et al., 2024) their reasoning trajectories and intermediate images during both inference and training. Motivated by this perspective, in this paper we investigate three central questions:

**Q1: What is the current performance of MLLMs on visual-outcome verification?**

To take a systematic assessment of MLLMs' current capabilities in verifying visual outcomes, we construct **ViVerBench**, a challenging and comprehensive benchmark that spans 16 subtasks across 6 categories of visual verification. The benchmark is meticulously constructed through manual annotation by 12 domain experts, resulting in 3,594 diverse and challenging verification questions. For each question, models are instructed to generate a binary true/false judgment accompanied by a detailed explanation, thereby enabling a fine-grained evaluation of both decision accuracy and reasoning correctness. We conduct experiments on 9 state-of-the-art VLMs and uncover 3 general limitations: (1) weakness in fine-grained and challenging image-prompt alignment, (2) mismatched representation of world knowledge, and (3) underdeveloped critics for visual reasoning tasks. These findings highlight a substantial gap between current VLMs and human-level visual verification, and indicate that there remains a long way to go before training VLM to become a powerful and truly multimodal universal verifier.

**Q2: How to develop a strong generative universal verifier?**

We investigate the potential of training a generative universal verifier by designing two automated data construction pipelines for visual verification, motivated by the idea of emulating human strategies for constructing challenging data. These pipelines focus on the alignment of fine-grained and challenging features in image-prompt alignment tasks, enabling the scalable creation of high-quality, diverse, and challenging training data. Leveraging this dataset, we directly applied reinforcement learning (RL) (Shao et al., 2024; Yu et al., 2025) to training Qwen2.5-VL-7B (Bai et al., 2025) under a rule-based verifier framework, resulting in **OmniVerifier-7B**. In ViVerBench, OmniVerifier-7B achieved an 8.3-point improvement and beat GPT-4o (Hurst et al., 2024). Furthermore, through ablation training on verification data across different tasks, we identify three fundamental atomic capabilities underlying visual-outcome verification: *explicit alignment*, *relational verification*, and *integrative reasoning*, and uncover how they mutually generalize and facilitate each other. These findings suggest a minimalist recipe for generative universal verifier training: *Instead of training for many tasks separately, constructing training data for fundamental atomic skills is sufficient to enable wide-ranging task generalization.*

**Q3: How can visual verification be leveraged to enhance reasoning or generation?**

Generative universal verifier admits a broad range of applications. We propose **OmniVerifier-TTS**, a sequential test-time scaling paradigm designed for enhancing the generation of unified multimodal models (Deng et al., 2025; Wu et al., 2025a) with OmniVerifier-7B. Starting from a generated image, it progressively refines images through multiple rounds of verification and editing, which bridging image generation and editing within a unified TTS framework. Validation across 2 advanced unified multimodal models demonstrates OmniVerifier-TTS achieves notable gains in general generative quality, including reasoning-based image generation (Sun et al., 2025), and complex compositional-based generation (Ye et al., 2025; Zhang et al., 2024c;b). We further demonstrate sequential TTS exhibits better performance compared to parallel TTS. Beyond TTS, we also extend the universal verifier to broader world-modeling scenarios, pushing the reasoning capabilities of MLLMs.

Our contributions can be summarized as:

- **Comprehensive and Challenging Benchmark for Visual-outcome Verification:** We introduce **ViVerBench**, a manually curated benchmark designed to evaluate visual-outcome verification in both multimodal reasoning and generation. Our evaluation reveals three general shortcomings of MLLMs.

- **Data Construction and Training Strategy for the Generative Universal Verifier:** We propose two automated data curation pipelines to scale visual verification training data, and train **OmniVerifier-7B**, achieving substantial improvements on ViVerBench and surpassing

GPT-4o. Our training identifies three atomic capabilities in visual-outcome verification and provides a minimalist recipe for training a generative universal verifier.

- **Sequential Test-time Scaling Paradigm of UMM:** We propose **OmniVerifier-TTS**, a flexible and general sequential test-time scaling paradigm for the generation of UMMs, suparssing other parallel TTS method such as Best-of-N while achieving a substantial reduction in inference time.

## 2 RELATED WORK

Recent breakthrough in multimodal large language models lie in in the continuously evolving reasoning paradigms (Peng et al., 2025; Shen et al., 2025; Zhang et al., 2025; Tong* et al., 2025) and more unified foundation models (Xie et al., 2025b; Chen et al., 2025a; Yang et al., 2025b; Guo* et al., 2025). Starting from simple image understanding, the long-chain-of-thought (longCoT) paradigm trained with reinforcement learning substantially strengthened multimodal reasoning, as seen in Seed-1.5VL (Guo et al., 2025), Kimi-VL(Team et al., 2025a). Scaling test-time reasoning over textual outcomes has become the mainstream strategy for improving reasoning ability. However, this text-centric approach treats vision as merely static context, leaving a semantic gap between perception and symbolic thought. Openai-o3 (OpenAI, 2025) brings visual outcomes into LongCoT, pioneering a new 'thinking with images' paradigm of interleaved multimodal reasoning. DeepEyes (Zheng et al., 2025) and MINT-CoT Chen* et al. (2025) demonstrate powerful tool-assisted and regsion-selected reasoning through end-to-end reinforcement learning. Meanwhile, architectures are converging toward unified designs that integrate text and image inputs/outputs (Chen et al., 2025c; Xie et al., 2025a), making interleaved reasoning within a single framework a compelling direction. Mogao (Liao et al., 2025) highlights the potential of interleaved generation under unified architectures, while T2I-R1 (Jiang et al., 2025) and Bagel (Deng et al., 2025) explore 'thinking before generating' for interleaved text–image synthesis. Looking ahead, we argue that self-critique will drive the next generation of multimodal reasoning by enabling autonomous learning through verification on both textual and visual outcomes, without reliance on external outcome labels.

## 3 VIVERBENCH: ASSESSMENT OF MLLMS ON VISUAL VERIFICATION

### 3.1 BENCHMARK OVERVIEW

**ViVerBench** is designed as a comprehensive and challenging benchmark to evaluate MLLMs' ability to verify visual outcomes during reasoning and generation. It comprises 16 tasks across 6 main categories: *Concept Existence*, *Object Relationship*, *World Dynamics*, *Image Annotation*, *State Value Evaluation*, and *STEM*, with representative examples shown in Fig. 1.

To ensure sufficient difficulty and unambiguous correctness, we construct the benchmark through a systematic pipeline that combines manual annotation, programmatic generation, and augmented open-source data. For evaluation, we further introduce two complementary metrics: rule-based and model-based, that assess not only answer accuracy but also explanation validity. Full task definitions, dataset construction details, and evaluation methodologies are provided in the Appendix A.

### 3.2 EVALUATION OF ADVANCED MLLMS

We evaluate a range of stat-of-the-art VLMs on **ViVerBench**, including closed-source models: Gemini-2.5-Pro (Comanici et al., 2025), GPT-5, GPT-4o (Hurst et al., 2024), OpenAI-o1/3/4mini (Jaech et al., 2024), Seed 1.5-VL (Guo et al., 2025), and two open-source models: Qwen2.5-VL 72B (Bai et al., 2025) and InternVL3.5 A28B (Wang et al., 2025a). We evaluate these models using rule-based verification, which directly determines the correctness of true/false labels without assessing the accompanying explanations using another judge model.

As shown in Table 1, Gemini-2.5-Pro achieves a state-of-the-art score of $0.745$ on **ViVerBench**; however, the score remains far from that of a reliable general visual verifier. In addition, we identify three reasons for the substantial gap between these advanced models and human evaluation:

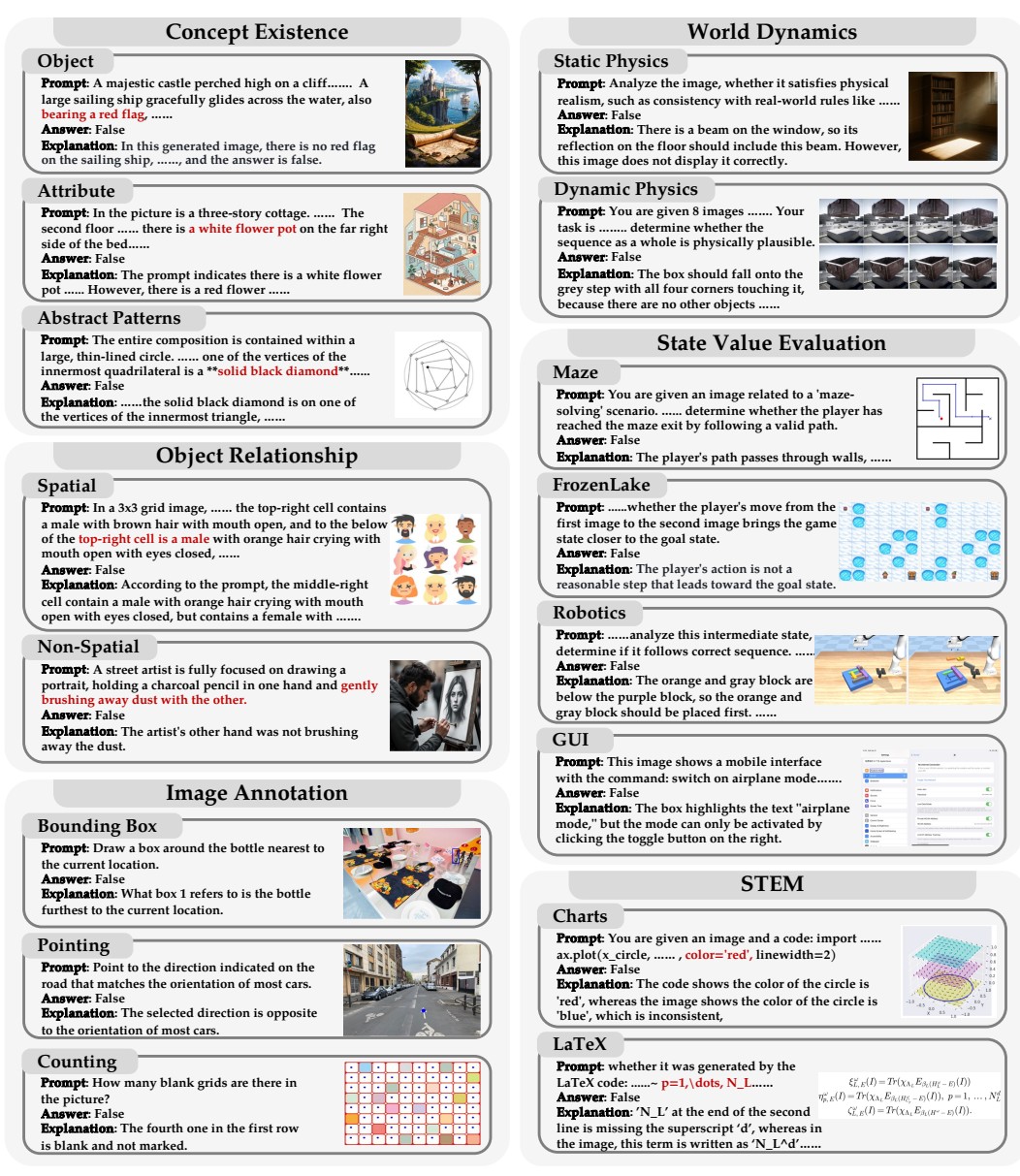

Figure 1: ViVerBench Overview. ViVerBench has a 1:1 ratio of true to false answers; here, we show only the false ones to better highlight data difficulty.

**Weakness in Fine-Grained and Challenging Image-Prompt Alignment**    State-of-the-art VLMs exhibit a performance gap of about $0.2$ compared to human performance on the three *Concept Existence* tasks. This shortfall stems from the difficulty models face in achieving precise, point-to-point alignment between complex compositional prompts and images, especially in cases of overlapping attributes across objects, small-scale elements, or occluded and blurred attributes. Human cognition employs deliberate and iterative verification strategies (i.e., 'double checking') to resolve such ambiguities, yielding far superior accuracy in visual verification.

**Mismatched Representation of World Knowledge**    Despite advanced VLMs equipped with extensive world knowledge, including general physical laws within their language components, our experiments reveal a significant paradox: this knowledge is not effectively activated in visual verification tasks. The notable performance gap between VLMs and humans in both *Static* and *Dynamic Physics* tasks reveals a fundamental Knowledge-Modality Gap.

Table 1: Rule-based evaluation of advanced VLMs on **ViVerBench**.

| Model / Metric | Concept Existence | | | Object Relation | | World Dynamics | | Image Annotation | | | State Value Evaluation | | | | STEM | | Overall |
|---|---|---|---|---|---|---|---|---|---|---|---|---|---|---|---|---|---|
| | Obj. | Attr. | Abs.P. | Spat. | N-Spat. | S-Phy | D-Phy | BBox | Point | Count | Maze | F.Lake | Robot. | GUI | Chart | LaTeX | |
| Qwen 2.5-VL 72B | 0.696 | 0.642 | 0.678 | 0.550 | 0.813 | 0.600 | 0.507 | 0.839 | 0.744 | 0.615 | 0.517 | 0.507 | 0.513 | 0.796 | 0.628 | **0.922** | 0.661 |
| InternVL3.5 A28B | 0.688 | 0.737 | 0.637 | 0.742 | 0.799 | 0.592 | 0.500 | 0.847 | 0.796 | 0.527 | 0.503 | 0.539 | 0.519 | 0.796 | 0.640 | 0.881 | 0.671 |
| GPT 4o | 0.540 | 0.608 | 0.671 | 0.538 | 0.731 | 0.713 | 0.500 | 0.649 | 0.744 | 0.632 | 0.570 | 0.643 | 0.563 | 0.796 | 0.656 | 0.758 | 0.645 |
| OpenAI o1 | 0.647 | 0.754 | 0.760 | 0.704 | 0.769 | 0.675 | 0.671 | 0.758 | 0.826 | 0.626 | 0.587 | 0.646 | 0.601 | 0.764 | 0.728 | 0.902 | 0.715 |
| OpenAI o4-mini | 0.745 | 0.746 | 0.781 | 0.763 | 0.754 | 0.646 | 0.654 | 0.843 | 0.819 | 0.604 | 0.560 | 0.650 | 0.658 | 0.833 | 0.700 | 0.876 | 0.727 |
| Seed 1.5-VL | 0.737 | **0.763** | 0.651 | 0.779 | **0.851** | 0.588 | 0.575 | **0.903** | 0.870 | 0.610 | 0.527 | 0.718 | **0.671** | 0.833 | 0.720 | 0.907 | 0.731 |
| OpenAI o3 | 0.723 | 0.728 | 0.801 | 0.713 | 0.754 | 0.729 | **0.682** | 0.802 | **0.885** | 0.643 | 0.517 | 0.671 | 0.627 | 0.875 | 0.732 | 0.887 | 0.735 |
| GPT-5 | 0.696 | 0.737 | 0.849 | 0.725 | 0.746 | **0.775** | 0.668 | 0.831 | **0.885** | 0.659 | 0.507 | 0.743 | 0.589 | 0.856 | **0.760** | 0.876 | 0.744 |
| Gemini 2.5 Pro | **0.763** | 0.750 | **0.856** | **0.875** | 0.761 | 0.746 | 0.532 | 0.875 | 0.863 | **0.698** | **0.580** | **0.804** | 0.563 | **0.912** | 0.540 | 0.799 | **0.745** |
| Random | 0.500 | 0.500 | 0.500 | 0.500 | 0.500 | 0.500 | 0.500 | 0.500 | 0.500 | 0.500 | 0.500 | 0.500 | 0.500 | 0.500 | 0.500 | 0.500 | 0.500 |
| Human | 0.938 | 0.940 | 0.932 | 0.988 | 0.955 | 0.929 | 0.818 | 0.961 | 0.966 | 0.918 | 0.997 | 1.000 | 1.000 | 0.935 | 0.928 | 0.706 | **0.932** |
| Qwen 2.5-VL 7B | 0.531 | 0.591 | 0.500 | 0.504 | 0.694 | 0.529 | 0.471 | 0.673 | 0.633 | 0.467 | 0.527 | 0.404 | 0.671 | 0.625 | 0.556 | 0.742 | 0.570 |
| OmniVerifier 7B (Ours) | 0.728 | 0.711 | 0.514 | 0.742 | 0.679 | 0.517 | 0.618 | 0.802 | 0.670 | 0.566 | 0.563 | 0.482 | 0.728 | 0.662 | 0.548 | 0.912 | 0.653 |

**Underdeveloped Critics for Visual Reasoning Tasks** The most significant performance gap between VLMs and human performance arises in tasks requiring reflective reasoning. In tasks like *Maze*, *FrozenLake*, and *Robotics*, humans achieve near-perfect performance, demonstrating robust reflective abilities. In contrast, most VLMs perform near chance level; for example, the best-performing model, Gemini 2.5 Pro, achieves only a score of $0.580$ on the *Maze* task. These results expose the inability of current models to reliably leverage learned rules for verification under complex visual reasoning scenarios.

> **Finding 1.** *Limitations of Advanced MLLMs in Visual Verification*
> - *Weakness in fine-grained and challenging image-prompt alignment*
> - *Mismatched representation of world knowledge*
> - *Underdeveloped critics for visual reasoning tasks*
>
> *These gaps highlight the distance from human-level visual verification.*

## 4 OmniVerifier: Building and Findings of Universal Verifier

The limited performance of current advanced MLLMs on visual verification motivates us to investigate methods that can progressively strengthen their capabilities. In Section 4.1, we introduce methods for scaling challenging visual verification data. Section 4.2 presents comprehensive ablation studies across diverse tasks to probe the core atomic capability of visual verifier, and Section 4.3 describes the performance of OmniVerifier-7B.

### 4.1 Automated Construction of Visual Verifier Data

We aim to scale both the quantity and quality of data through automated construction rather than heavy manual effort. The core challenge is **avoiding overly nitpicking judgments**. Current generative models still fall short of fully aligning with complex prompts, leaving room for subjective criticism. Specifically, they often fail to maintain precise object attributes and consistent spatial relationships in complex scenes. Prompts also often involve subjective or abstract notions (e.g., atmosphere, style, fine-grained details), which lack uniquely correct answers and thus make images seem 'not fully aligned.' This also explains why advanced MLLMs struggle with fine-grained and challenging image–prompt alignment, therefore, the key lies in the construction of high-quality, challenging, and rigorously accurate visual verification data.

We begin with complex images and construct true and false examples in a reversed manner. For synthetic images, prompts are drawn from ShareGPT-4o-Image (Chen et al., 2025b) and further enriched by GPT-5, with a focus on introducing multiple objects, diverse attributes, and both spatial and non-spatial relationships. These complex prompts are then used with Seedream 3.0 (Gao et al., 2025) to generate complex images. For natural images, 20k samples are taken from LVIS (Gupta et al., 2019), and GPT-5 is applied to filter out simpler ones, retaining only complex cases. This process yields a repository of complex images, which serves as the foundation for subsequent visual verifier data construction. To achieve optimal generalization and data augmentation effects, we design two automated pipelines shown in Fig. 2 for paired true and false data construction:

**Method1: Image-Fixed, Prompt-Modified** We construct true and false data by modifying prompts. For each complex image, we first use GPT-5 to generate a strictly constrained prompt that

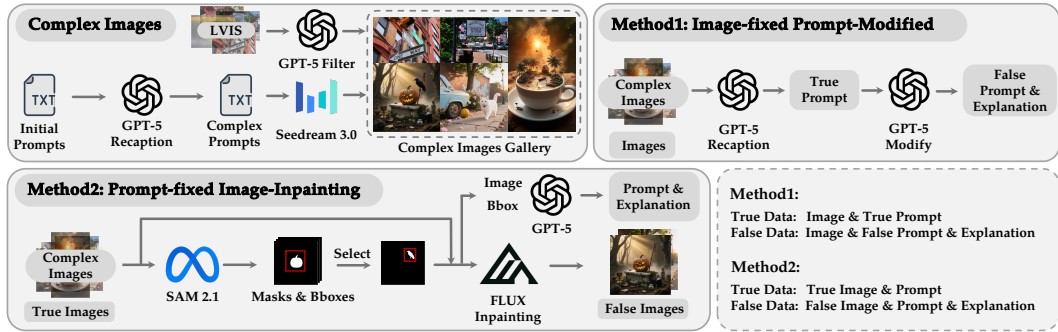

Figure 2: Automated pipeline for visual verifier data construction.

describes only clearly identifiable elements (objects, attributes, spatial relationships, and scenes), while avoiding speculation, subjective judgments, or unnecessary embellishment. This yields highly faithful prompts that align closely with the complex image, which we treat as true data. We then further modify these prompts with GPT-5 by altering details, for example, adding or removing objects, changing attributes, or modifying spatial relations and provide corresponding explanations.

**Method2: Prompt-Fixed, Image-Inpainting** We construct true and false data by inpainting images. We first apply SAM 2.1 (Ravi et al., 2024) to segment all objects in the complex image, obtaining the corresponding masks and bounding boxes. The mask area serves as a measure of data difficulty, based on which we dynamically select masks and use FLUX.1-dev (Labs, 2024) for inpainting to generate false images. Meanwhile, GPT-5 is employed to generate strictly constrained prompts, where the bounding box of selected object is highlighted to ensure explicit descriptions of its attributes and spatial position. This design prevents incomplete prompts in high-difficulty cases. Finally, we obtain both the prompts and their accompanying explanations.

Following the construction of visual verifier data via Method 1 and Method 2, we apply Seed1.5-VL (Guo et al., 2025) to clean and retain only samples with a Best-of-10 accuracy of at least 0.6.

## 4.2 GENERALIZATION OF ATOMIC CAPABILITIES IN THE VISUAL VERIFIER

With these high-quality, scalable data, we can more effectively investigate the essence of visual verifiers. Although ViVerBench covers a wide range of tasks, it remains unclear whether these tasks are intrinsically connected. To investigate these potential connections and to explore how to train a more comprehensive visual verifier, we focus on four fundamentally distinct tasks. Specifically, we construct object and attribute datasets following the approach in Section 4.1, and additionally include in-domain spatial and maze datasets. We select these tasks because each evaluates a complementary aspect of visual verification: object and attribute tasks probe the foundational ability of explicit image-text alignment, the spatial task captures more complex relational reasoning beyond basic alignment, and the maze task serves as a reasoning challenge, assessing visual verification in the reasoning dimension. Verifier training is conducted independently on each dataset, resulting in four distinct experimental settings.

We apply DAPO (Yu et al., 2025) to perform RL training directly on Qwen2.5-VL-7B (Bai et al., 2025), using a system prompt that encourages the model to reason before answering. The training objective combines a rule-based reward, which evaluates the correctness of true/false predictions, with a format reward at a 9:1 ratio. All four models are trained for 100 steps on 64 NVIDIA A100-80G GPUs, and the results are shown in Fig 3.

Training only on object verification data yields significant improvements across most tasks. For tasks require explicit prompt-image alignment, such as *Attribute*, *Charts*, and *LaTeX*, traing on object verification data provides notable gains. It also generalizes well to tasks that involve verifying relationships between objects in the prompt, including *Spatial*, *Static Physics*, *Bounding Box*, *Pointing*, and *GUI*. However, for visual reasoning tasks like *Maze* and *Robotics*, performance remains instability and does not yield noticeable gains. Notably, the generalization trend of attribute verification data shows a similar trend to object verification data.

Training only on spatial verification data also shows a strong generalization to explicit alignment tasks such as *Object*, *Attribute*, *Charts*, and *LaTeX*. Moreover, it delivers even larger improvements

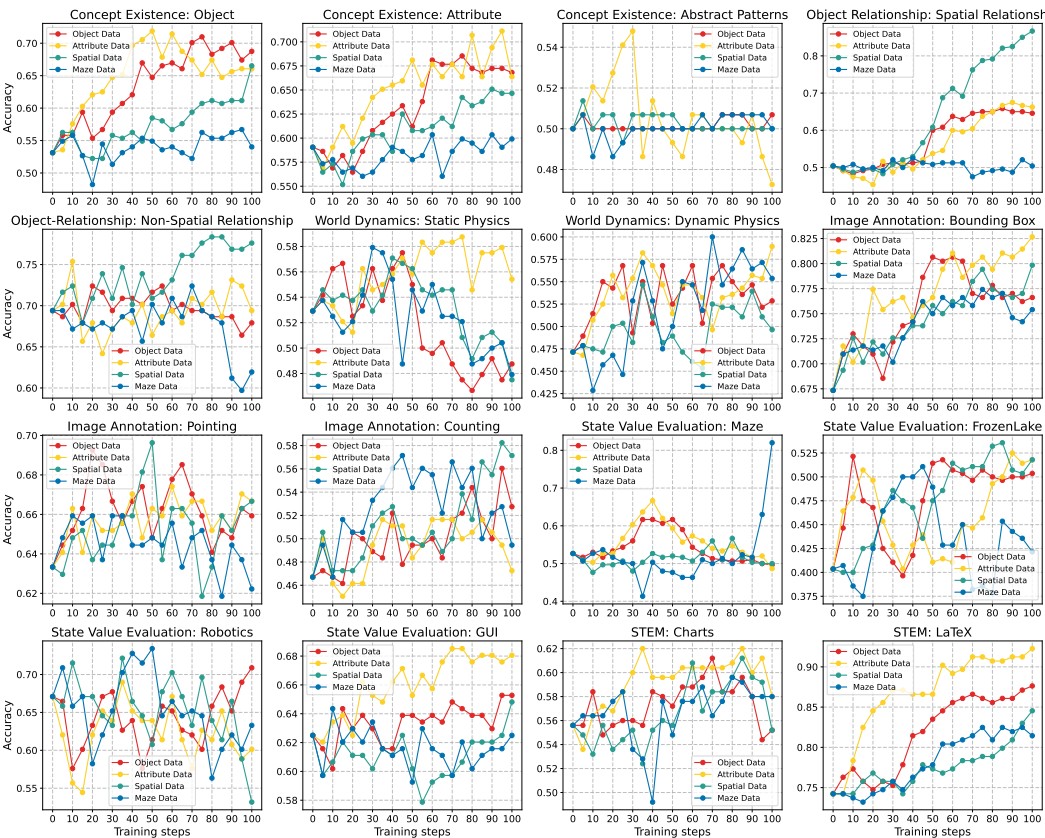

Figure 3: Training progress of four visual verification tasks on ViVerBench.

on relational tasks, including *Non-Spatial*, *Bounding Box*, and *Counting*. Nevertheless, for complex visual reasoning tasks like *Maze* and *Robotics*, the benefits remain limited.

By contrast, training only on maze verification data exhibits minimal generalization. We attribute this to the sparse and discrete nature of maze images, where paths are rendered as blank space and walls as simple black lines. Such simplistic, synthetic patterns contrast sharply with the rich textures and semantics of natural images, creating a significant distribution gap. As a result, maze data offers limited transferable signal, and we observe no meaningful gains on broader tasks.

**Finding 2.** *Three Progressively Related Atomic Capabilities in Visual Verification*

- **Explicit Alignment**: *text and image contain directly matchable elements.*
- **Relational Verification**: *requires using text to verify relationships or perform light reasoning, beyond simple visual matching.*
- **Integrative Reasoning**: *involves holistic interaction between prompt and image for complex or higher-order reasoning.*

*These capabilities form a layered structure, from perceptual to semantic-relational, and finally to task-level reasoning.*

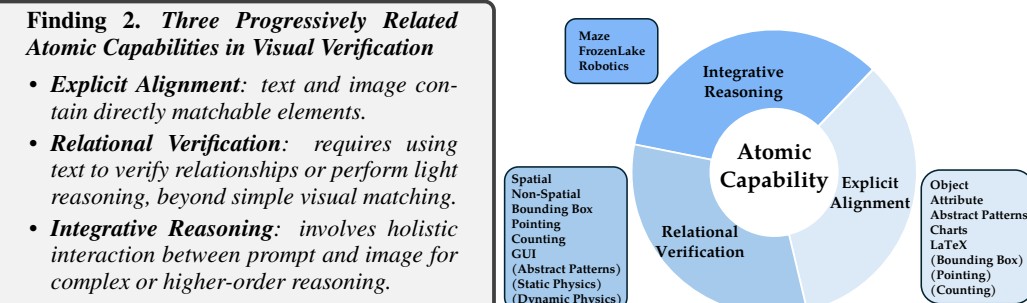

Figure 4: Atomic capabilities in visual verifier.

We categorize the tasks associated with the three atomic capabilities of visual verification, as illustrated in Fig 4. Tasks shown in parentheses indicate cases where the underlying atomic capability may shift with prompt complexity, rather than being fixed to a single type. Our experiments also demonstrate that reinforcement learning facilitates the generalization across atomic capabilities (Yuan et al., 2025): we observe strong mutual improvment both within and between *Explicit*

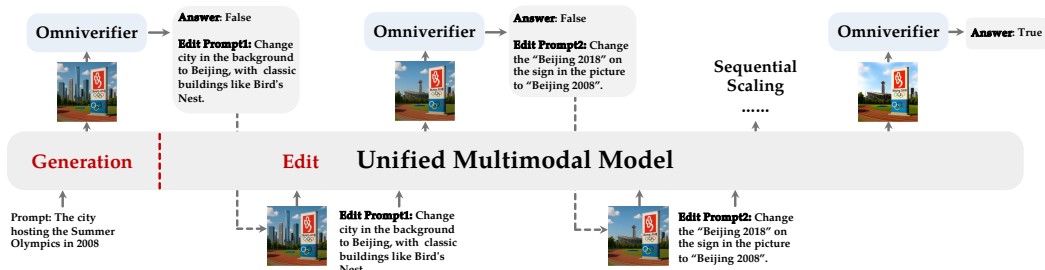

Figure 5: OmniVerifier-TTS: sequential test-time scaling pipeline of unified multimodal models.

*Alignment* and *Relational Verification*. This yields an important training insight for building a universal verifier: **it is unnecessary to construct task-specific datasets. Instead, a single dataset that captures the underlying visual patterns of these two atomic capabilities is sufficient to enable broad cross-task generalization.**

In contrast, *Integrative Reasoning* poses a fundamentally different challenge. Tasks in this category often span highly diverse domains, where visual inputs, reasoning patterns, and solution strategies differ substantially. These pronounced domain gaps make it difficult to establish shared representations, and training on one task task yields minimal transfer to others. Accordingly, we recommend building task-specific datasets tailored to each domain to effectively improve integrative reasoning.

> **Finding 3.** *Generalization of Three Visual Verification Atomic Capabilities*
> *Reinforcement learning promotes strong generalization within and between Explicit Alignment and Relational Verification, suggesting that a single dataset capturing their shared visual patterns suffices for broad transfer. In contrast, Integrative Reasoning spans heterogeneous domains with little cross-task transfer, thus requiring task-specific datasets.*

### 4.3 OMNI-CAPABLE GENERATIVE UNIVERSAL VERIFIER

The findings on atomic capabilities and their generalization motivate training a comprehensive generative universal verifier. Following Methods 1 and 2 in Section 4.1, we construct 28k high-quality visual verification datasets, filtered with Seed1.5-VL and covering both *Explicit Alignment* and *Relational Verification*. Using the training procedure in Section 4.2 with Qwen2.5-VL-7B as the backbone, we obtain **OmniVerifier-7B**. Its performance on ViVerBench is reported in Table 1.

With high-quality atomic-level data, reinforcement learning training has endowed the base model with exceptionally potent visual verification abilities. OmniVerifier-7B demonstrates a significant 8.3% overall performance enhancement, surpassing GPT-4o and achieving capabilities comparable to Qwen2.5-VL-72B. Notably, OmniVerifier-7B exhibits pronounced improvements in tasks related to *Explicit Alignment* and *Relational Verification*, such as *Object*, *Attribute*, *Spatial* and *Bounding Box*. Our findings suggest that targeted reinforcement learning on atomic capabilities offers a promising direction for building stronger and more generalizable visul verifiers.

## 5 OMNIVERIFIER-TTS: MULTIMODAL SEQUENTIAL TEST-TIME SCALING

In this section, we aim to explore how to apply the universal verifier to multimodal generation and reasoning. Specifically, we present **OmniVerifier-TTS**, which integrates image generation and editing within unified multimodal models through sequential test-time scaling, enabling a flexible paradigm of interleaved generation. Section 5.1 describes the detailed structure of the framework; Section 5.2 presents experiments and visualizations of OmniVerifier-TTS; and Section 5.3 compares parallel and sequential test-time scaling in UMMs.

### 5.1 ARCHITECTURE

Achieving precise image generation from complex compositional or reasoning-based prompts is often difficult in a single attempt. However, in many cases, only a small region or small parts of the image is misaligned with input prompt, making it unnecessary to regenerate the entire image or

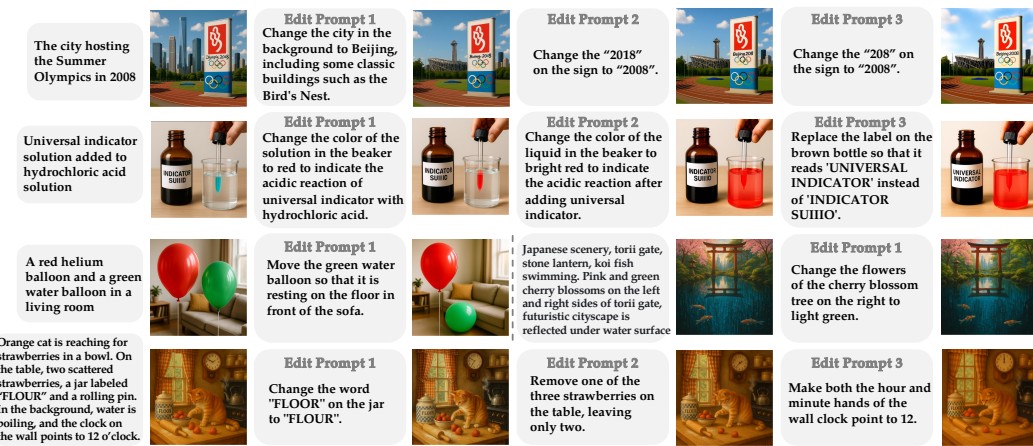

Figure 6: Qualitative visualization of OmniVerifier-TTS.

introduce large changes. A more optimal solution is to perform targeted, regional edits to correct these minor errors and obtain a highly-aligned result.

Motivated by this, we propose a self-refinement pipeline that leverages visual verification to identify inconsistencies. As illustrated in Fig. 5, we employ OmniVerifier as a 'misalignment-finder' due to its strong capability in explicit alignment and relational verification. The process begins with UMM generating an image from a given prompt. OmniVerifier then analyzes this image and outputs a binary judgment (true/false) along with an explanation, following the same procedure as in its RL training. If the judgment is false, indicating misalignment between the prompt and the image, OmniVerifier additionally outputs an edit prompt, a rephrased form of the explanation that offers instructive guidance on how the image should be modified. The UMM subsequently performs fine-grained edits on the initial image based on this edit prompt to obtain a refined result. This iterative refinement loop continues OmniVerifier returns a true judgment or the maximum number of refinement steps is reached.

## 5.2 EXPERIMENT

We conduct experiments using two powerful open- and closed-source models, Qwen-Image (Wu et al., 2025a) and GPT-Image-1, with OmniVerifier-7B serving as the judge model. The maximum number of refinement steps is set to 10. All experiments are conducted on a single NVIDIA-A100-80G GPU. We evaluate on a reasoning-based benchmarks, T2I-ReasonBench (Sun et al., 2025), as well as one complex compositional-based benchmark, GenEval++ (Ye et al., 2025). The results are presented in Table 2.

Table 2: Evaluation of **OmniVerifier-TTS** on reasoning and compositional generation benchmarks.

| Model / Metric | Reason-based Generation | | | | | Compositional-based Generation | | | | | | | |
|---|---|---|---|---|---|---|---|---|---|---|---|---|---|
| | T2I-ReasonBench | | | | | GenEval++ | | | | | | | |
| | Idiom | Textual | Entity | Scientific | Overall | Color | Count | Color/Count | Color/Pos | Pos/Count | Pos/Size | Multi-Count | Overall |
| **SD-3-Medium** (Esser et al., 2024) | 35.9 | 60.9 | 42.4 | 50.9 | 47.5 | 0.550 | 0.500 | 0.125 | 0.350 | 0.175 | 0.150 | 0.225 | 0.296 |
| **FLUX.1-dev** (Labs, 2024) | 39.1 | 56.9 | 45.1 | 46.7 | 47.0 | 0.350 | 0.625 | 0.150 | 0.275 | 0.200 | 0.375 | 0.225 | 0.314 |
| **Janus-Pro** (Chen et al., 2025c) | 25.5 | 37.2 | 38.5 | 44.9 | 36.5 | 0.450 | 0.300 | 0.125 | 0.300 | 0.075 | 0.350 | 0.125 | 0.246 |
| **Bagel** (Deng et al., 2025) | 44.6 | 44.0 | 52.4 | 57.7 | 49.7 | 0.325 | 0.600 | 0.250 | 0.325 | 0.250 | 0.475 | 0.375 | 0.371 |
| **Qwen-Image** (Wu et al., 2025a) | 46.5 | 66.3 | 53.4 | 55.8 | 55.5 | 0.700 | 0.900 | 0.800 | 0.525 | 0.500 | 0.700 | 0.600 | 0.675 |
| **QwenVL-TTS(Qwen-Image)** | 50.1 | 67.1 | 55.9 | 56.5 | 57.4 | 0.750 | 0.850 | 0.825 | 0.550 | 0.500 | 0.675 | 0.625 | 0.682 |
| **OmniVerifier-TTS(Qwen-Image)** | 51.1 | 68.4 | 58.5 | 58.7 | 59.2 | 0.800 | 0.925 | 0.825 | 0.600 | 0.525 | 0.700 | 0.650 | 0.718 |
| **GPT-Image-1** | 75.4 | 84.6 | 75.7 | 71.6 | 76.8 | 0.650 | 0.800 | 0.750 | 0.575 | 0.550 | 0.725 | 0.775 | 0.689 |
| **QwenVL-TTS(GPT-Image-1)** | 76.0 | 86.6 | 76.4 | 72.5 | 77.8 | 0.675 | 0.800 | 0.700 | 0.575 | 0.550 | 0.750 | 0.800 | 0.693 |
| **OmniVerifier-TTS(GPT-Image-1)** | 78.1 | 87.4 | 77.8 | 73.7 | 79.3 | 0.725 | 0.825 | 0.750 | 0.600 | 0.575 | 0.750 | 0.825 | 0.721 |

OmniVerifier-TTS brings multifaceted improvements to T2I generation. In reason-based generation, it achieves substantial gains on T2I-ReasonBench, boosting Qwen-Image by 3.7 points and GPT-Image-1 by 2.5 points through sequential TTS refinement. In complex composition-based generation, OmniVerifier-TTS demonstrates clear advantages in aspects such as Color and Pos/Count. These benefits stem from OmniVerifier-TTS's architecture, which continuously refines generated images within small-scale adjustments and leverages a generative verifier to combine generation and editing capabilities. Additionally, using Qwen2.5-VL-7B as the verifier, QwenVL-TTS lags

behind OmniVerifier-TTS on both benchmarks, underscoring OmniVerifier's superior and accurate visual verification ability in the general T2I domain.

OmniVerifier-TTS unlocks potential for interleaved image-text generation and reasoning. As shown in Fig. 6, for complex T2I tasks, OmniVerifier progressively generates semantically consistent and high-quality images through iterative self-refinement. It can precisely identify and correct unrealistic or flawed elements in images, particularly those related to physics or authenticity. This reasoning paradigm enables automated scaling of interleaved image-text data generation. We firmly believe that OmniVerifier-TTS not only enhances image generation but also provides a robust data infrastructure and performance guarantee for the forthcoming era of interleaved image-text systems.

## 5.3 COMPARISION BETWEEN PARALLEL AND SEQUENTIAL TEST-TIME SCALING

To further validate the advantages of sequential OmniVerifier-TTS, we also explore alternative parallel test-time scaling strategies, such as Best-of-N shown in Fig 7. Specifically, OmniVerifier-7B is employed to compare $N$ images generated from the same prompt through progressive pairwise selection, where the final winner is chosen as the best image. We compare the sequential and parallel OmniVerifier-TTS on Qwen-Image and GPT-Image-1 with $N = 10$, and the results are reported in Table 3.

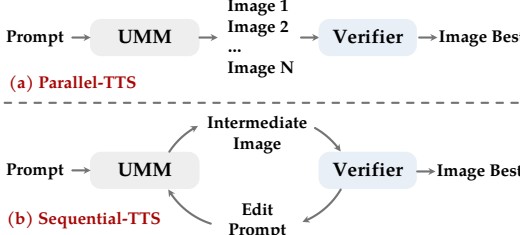

Figure 7: Comparision between Parallel and Sequential Test-Time Scaling.

Table 3: Evaluation of Parallel and Sequential Test-Time Scaling in UMMs.

| Model / Metric | Reason-based Generation | | | | | Compositional-based Generation | | | | | | | |
| --- | --- | --- | --- | --- | --- | --- | --- | --- | --- | --- | --- | --- | --- |
| | T2I-ReasonBench | | | | | GenEval++ | | | | | | | |
| | Idiom | Textual | Entity | Scientific | Overall | Color | Count | Color/Count | Color/Pos | Pos/Count | Pos/Size | Multi-Count | Overall |
| Qwen-Image (Wu et al., 2025a) | 46.5 | 66.3 | 53.4 | 55.8 | 55.5 | 0.700 | 0.900 | 0.800 | 0.525 | 0.500 | 0.700 | 0.600 | 0.675 |
| OmniVerifier-TTS(Parallel) | 49.3 | 68.8 | 56.2 | 58.2 | 58.1 | 0.750 | 0.900 | 0.775 | 0.575 | 0.500 | 0.700 | 0.650 | 0.693 |
| OmniVerifier-TTS(Sequential) | 51.1 | 68.4 | 58.5 | 58.7 | 59.2 | 0.800 | 0.925 | 0.825 | 0.600 | 0.525 | 0.700 | 0.650 | 0.718 |
| GPT-Image-1 | 75.4 | 84.6 | 75.7 | 71.6 | 76.8 | 0.650 | 0.800 | 0.750 | 0.575 | 0.550 | 0.725 | 0.775 | 0.689 |
| OmniVerifier-TTS(Parallel) | 77.0 | 85.4 | 77.4 | 72.4 | 78.1 | 0.700 | 0.825 | 0.750 | 0.575 | 0.550 | 0.725 | 0.775 | 0.700 |
| OmniVerifier-TTS(Sequential) | 78.1 | 87.4 | 77.8 | 73.7 | 79.3 | 0.725 | 0.825 | 0.750 | 0.600 | 0.575 | 0.750 | 0.825 | 0.721 |

Sequential TTS offers a higher performance ceiling than Parallel TTS. As shown in Table 3, it consistently outperforms Parallel TTS across all three benchmarks. Moreover, it requires fewer generation steps per prompt: while Parallel TTS generates 10 images per prompt, Sequential TTS achieves superior results in approximately 47% of the time. This efficiency and advanced performance arises from its ability to fully exploit the generative critiques provided by OmniVerifier, enabling multi-round, fine-grined optimization through UMM.

---

**Finding 4.** *Advantages of Sequential OmniVerifier-TTS*
*Equipped with generative verifier, sequential TTS has a higher performance ceiling than Parallel TTS in unified multimodal models.*

---

## 6 CONCLUSION

This paper focuses on *Generative Universal Verifier* and makes three key contributions. First, we introduce **ViVerBench**, which to the best of our knowledge, is the first comprehensive benchmark for evaluating MLLMs' verification of visual outcomes. Second, we develop two automated data construction pipelines and explore effective training strategies for universal verifiers, culminating in the powerful **OmniVerifier-7B**. Third, we investigate practical applications of universal verifiers, proposing **OmniVerifier-TTS**, a sequential test-time scaling method that enhances image generation, and further exploring broader world modeling reasoning scenarios. In future work, we aim to scale up the universal verifier and examine its potential for improving multimodal post-training.

## ACKNOWLEDGEMENT

This work was partly supported by the National Natural Science Foundation of China (Grant No. 62576191) and the Shenzhen Science and Technology Program (ZDCY20250901103533010) .

## ETHICS STATEMENT

This paper advances the field of multimodal large language models, while emphasizing the importance of responsible use to avoid potential negative societal impacts, such as the creation of misleading or harmful content.

## REPRODUCIBILITY STATEMENT

Our code is included in the supplementary material, and the implementation details are described in Section 4.2.

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

# A  DETAILS OF VIVERBENCH

## A.1  TASK DEFINITION

**Concept Existence**   evaluates whether all elements described in the prompt are accurately represented in the image, particularly in complex text-image alignment tasks such as compositional text-to-image generation. This evaluation consists of the following components:

- **Object:** Assesses whether all objects mentioned in the prompt are present in the image. Challenges arise in complex scenarios, such as detecting small objects, distinguishing between objects sharining overlapping attributes or easily confusable objects, and identifying occluded objects.

> **Illustrative Example of Object Data**
>
> 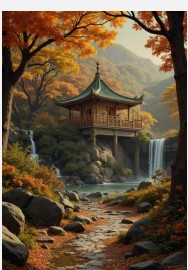
>
> **Question:** This image was generated from the prompt: "This stunning image captures a serene autumn scene featuring a traditional East Asian-style pavilion, possibly a temple, majestically perched on a rocky outcrop beside a tranquil pool. The spire of another temple is faintly visible behind it. Waterfalls cascade into the water, enhancing the peaceful atmosphere. The surrounding trees are ablaze with vibrant orange and yellow foliage, indicating the beauty of autumn. A stone path strewn with fallen leaves leads towards the pavilion. "
> Please carefully analyze the image and determine whether all the objects and their quantities mentioned in the prompt are correctly represented in the image. If all the objects and quantities are correctly presented, please answer 'true'; otherwise, answer 'false'.
> Provide your evaluation strictly in the following JSON format:
> { "answer": true/false, "explanation": "If the answer is false, briefly summarize the main error."
> }
> **Answer:** False
> **Explanation:**   In this generated image, the spire of another pavilion cannot be seen behind the central pavilion, so that object is missing, and the answer is false.

- **Attribute:** Evaluates whether all attributes such as color, quantity, and expression specified in the prompt are correctly depicted. The difficulty increases when multiple attributes must be satisfied simultaneously, and when ensuring accurate binding of attributes to the correct objects in complex scenes.

> **Illustrative Example of Attribute Data**
>
> 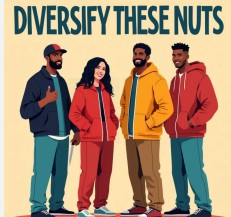
>
> **Question:** This image was generated from the prompt: "This image features four diverse individuals standing in a row against a plain, light background. Above them, bold, dark blue text reads "DIVERSIFY THESE NUTS." From left to right, the first man wears a dark blue jacket

over a lighter shirt, teal pants, and a baseball cap. The woman in the center has dark, wavy hair and is dressed in a yellow jacket and red pants. To her right, a man with a beard smiles in a yellow hoodie and brown pants. The last man on the right sports a red hoodie and teal pants. All four appear relaxed and friendly, with subtle variations in their styles."

Please carefully analyze the image and determine whether all the attributes specified in the prompt (such as color, texture, shape, material, lighting, expression, and motion) are correctly represented in the image. If all the attributes are correctly presented, please answer 'true'; otherwise, answer 'false'.

Provide your evaluation strictly in the following JSON format:
{ "answer": true/false, "explanation": "If the answer is false, briefly summarize the main error."
}

**Answer:** False

**Explanation:** In the generated image, the woman's jacket color is red, not yellow; therefore, the color attribute of this jacket is incorrect, and the answer is false.

- **Abstract Patterns:** Focuses on high-level verification tasks involving visual logic puzzles with abstract, challenging patterns and plausible distractors. It tests the model's ability to critic about compositional relationships among multiple objects and attributes from varied perspectives within abstract scenarios.

---

Illustrative Example of Abstract Patterns Data

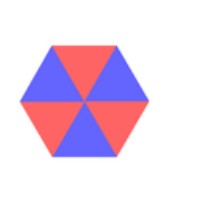

**Question:** This image was generated by from this prompt: Here is a detailed description of the figure: This is a digital graphic featuring a highly symmetrical, geometric design set against a plain white background. **Overall Shape and Composition:** The main figure is a regular hexagon, oriented with a flat top and bottom. This hexagon is perfectly subdivided into six identical equilateral triangles. The vertices of all six triangles meet at the precise center of the hexagon, creating a star-like junction. The base of each triangle corresponds to one of the six outer sides of the hexagon. **Color and Pattern:** The design employs a simple two-color palette: * A warm, vibrant coral or salmon-pink. * A cool, rich shade of blue, similar to periwinkle or royal blue. These two colors are applied to the triangles in a strict alternating pattern. Moving clockwise (or counter-clockwise) around the center, the colors cycle between pink and blue. This results in four pink triangles and two blue triangles. No two triangles of the same color are adjacent. **Visual Effect and Style:** * **Symmetry:** The figure possesses 3-fold rotational symmetry (C3), meaning it looks identical if rotated by 120 or 240 degrees. The alternating color scheme gives it a dynamic, pinwheel-like appearance. * **Style:** The style is minimalist and flat. There are no gradients, shadows, or textures—just solid blocks of color with clean, sharp edges. * **Analogy:** The pattern is reminiscent of a kaleidoscope, a spinning top, the top-down view of a segmented umbrella, or a faceted gem. The contrast between the warm pink and cool blue makes the design visually engaging and balanced.

Please carefully analyze the image and determine whether the generated image strictly matches the prompt, including aspects such as position, color, quantity, and shape. Answer 'true' if it does, and 'false' if it doesn't.

Provide your evaluation strictly in the following JSON format:
{ "answer": true/false, "explanation": "If the answer is false, briefly summarize the main error."
}

**Answer:** False

**Explanation:** The description states that the design results in four pink triangles and two blue triangles. However, in the image, there are three pink triangles and three blue triangles. This contradiction indicates that the description does not match the image.

**Object Relationship**    evaluates the model's ability to verify the spatial relationship and interaction between objects in complex sceniors. This evaluation consists of the following components:

- **Spatial:** Evaluates the model's capability to detect spatial misalignments between objects in images and the corresponding prompts, especially under varying viewpoints and in complex scenes that combine positional relationships with multiple object attributes.

> **Illustrative Example of Spatial Data**
>
> 
>
> **Question:** This image was generated from the prompt: In a 3x3 grid image, the bottom-left cell contains a female with orange hair with eyes closed, and to the top-right of the bottom-left cell is a female with purple hair with mouth open, the top-right cell contains a male with brown hair with mouth open, and to the below of the top-right cell is a male with orange hair crying with mouth open with eyes closed, and to the bottom-left of the middle-right cell is a female with yellow hair with eyes closed.
> Please carefully analyze the image and determine whether the spatial relationships between objects mentioned in the prompt are correctly represented in the image. If all the spatial relationships are correctly presented, please answer 'true'; otherwise, answer 'false'.
> Provide your evaluation strictly in the following JSON format:
> { "answer": true/false, "explanation": "If the answer is false, briefly summarize the main error."
> }
> **Answer:** False
> **Explanation:** The spatial relationship is incorrect. According to the prompt, the middle-right cell should contain a male with orange hair crying with mouth open with eyes closed, but instead contains a female with pastel pink and blonde hair with eyes closed. Therefore, the image does not match the prompt.

- **Non-Spatial:** Focuses on interactions between objects that cannot be easily characterized by spatial positioning alone (Huang et al., 2023), with particular attention to challenging cases involving complex interactions between a single object and several others.

> **Illustrative Example of Non-Spatial Data**
>
> 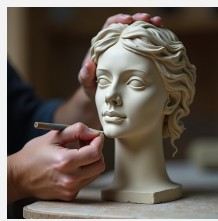
>
> **Question:** This image was generated from the prompt: "A sculptor is carving intricate details into a clay bust, shaping with one hand while supporting the sculpture's chin with the other."
> Please carefully analyze all elements in the prompt that involve interactions between two objects or actions involving people, and check whether all these relationship-related words are correctly reflected in the generated image. If all such relationships are accurately depicted, please answer 'true'; otherwise, answer 'false'.
> Provide your evaluation strictly in the following JSON format:
> { "answer": true/false, "explanation": "If the answer is false, briefly summarize the main error."
> }
> **Answer:** False
> **Explanation:** The sculptor's hand is not holding the sculpture's chin.

**World Dynamics** evaluates whether a single image or a sequence of images conforms to real-world physical laws, examining the model's world knowledge and its ability to judge physical plausibility in visual domain. This evaluation consists of the following components:

- **Static Physics:** Assesses whether a single generated image strictly obeys physical laws such as gravity, buoyancy, and lighting effects (e.g., shadows and reflections). To increase difficulty, the constructed data contain subtle yet clearly identifiable violations.

> **Illustrative Example of Static Physics Data**
>
> 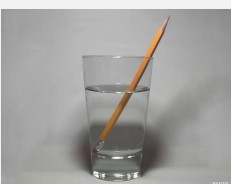
>
> **Question:** This image was generated by a model. Please carefully analyze the image and determine whether it satisfies physical realism, such as consistency with real-world rules like lighting, gravity, melting points, and other physical laws.
> Answer 'true' if it does, and 'false' if it doesn't.
> Provide your evaluation strictly in the following JSON format:
> { "answer": true/false, "explanation": "If the answer is false, briefly summarize the main error."
> }
> **Answer:** False
> **Explanation:** A pencil is placed in a water cup filled with water. The part of the pencil under the water should be offset to the right.

- **Dynamic Physics:** Evaluates whether a sequence of images follows consistent physical laws over time, assessing the model's ability to judge temporal dynamics and physical plausibility in dynamic scenarios.

> **Illustrative Example of Dynamic Physics Data**
>
> 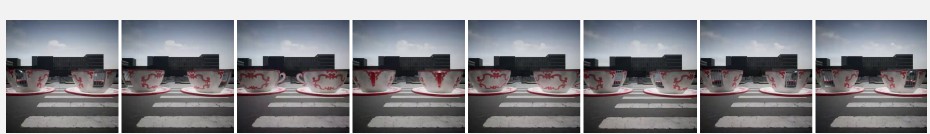
>
> **Question:** You are given 8 consecutive images representing a physical event or motion. Your task is to analyze the temporal progression and determine whether the sequence as a whole is physically plausible. Assume the event is a closed system, unfolding naturally without any unseen external force or human intervention. Consider a wide range of physical principles, including: - Object Permanence: Objects should not appear or disappear without a physical cause. - Continuity of Motion: The movement of objects should be smooth and logical. - Consistent Positions & Interactions: Objects should interact with each other and their environment in a consistent manner. - Plausible Dynamics: Any acceleration, deformation, or change in state must align with real-world physics (e.g., gravity, momentum). - Causal Relationships: The state of the scene in one frame should be a direct and logical cause of the state in the next.
> Answer 'true' if the sequence as a whole appears physically realistic, otherwise answer 'false'.
> Provide your evaluation strictly in the following JSON format:
> { "answer": true/false, "explanation": "If the answer is false, briefly summarize the main error."
> }
> **Answer:** False
> **Explanation:** The ball in the left should not appear in the right.

**Image Annotation** evaluates whether image annotation outputs accurately satisfy the requirements specified in the prompt. This evaluation consists of the following components:

- **Bounding Box:** Assesses a model's ability to judge whether a provided bounding box correctly corresponds to the specific object indicated in the prompt, with particular emphasis on objects that have ambiguous attributes or complex spatial configurations.

> **Illustrative Example of Bounding Box Data**
>
> 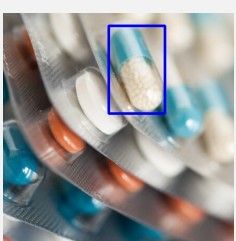
>
> **Question:** You are give an image with a blue bounding box indicating a selected region to solve the question: Draw a box around the white, circular pill. Evaluate whether the blue bounding box shown on the image accurately point out the correct answer. Note: All positional descriptions are given from the photographer's perspective.
> Answer 'true' if the blue bounding box accurately point out the correct answer, otherwise answer 'false'.
> Provide your evaluation strictly in the following JSON format:
> { "answer": true/false, "explanation": "If the answer is false, briefly summarize the main error."
> }
> **Answer:** False
> **Explanation:** The selected area is translucent capsule filled with white particles, not white, circular pill.

- **Pointing:** Similar to Bounding Box, this evaluates a model's ability to judge the correctness of provided points for specific objects in the image.

> **Illustrative Example of Pointing Data**
>
> 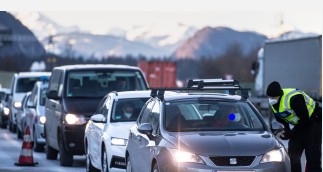
>
> **Question:** You are give an image marking with blue point to a pointing task, which solves the question: Point to the person speaking to someone in the car. Evaluate whether the blue points shown on the image accurately point out the correct answer. Note: All positional descriptions are given from the photographer's perspective.
> Answer 'true' if the blue point accurately point out the correct answer, otherwise answer 'false'.
> Provide your evaluation strictly in the following JSON format:
> { "answer": true/false, "explanation": "If the answer is false, briefly summarize the main error."
> }
> **Answer:** False
> **Explanation:** Point 1 refers to a person who is sitting in a car and talking to someone outside.

- **Counting:** Measures whether all objects specified in the prompt are correctly represented by points in the image. Each point must correspond to the correct object, and no specified objects should be missed, making this task more challenging than standard pointing tasks.

---

**Illustrative Example of Counting Data**

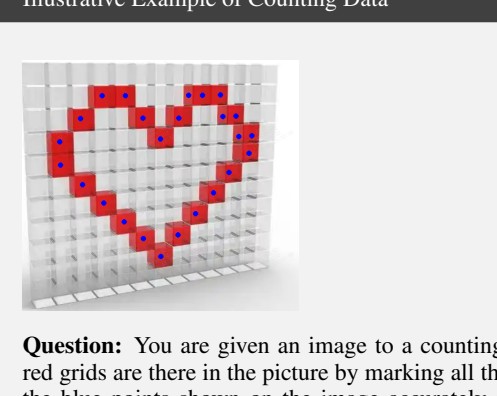

**Question:** You are given an image to a counting task, which solves the question: How many red grids are there in the picture by marking all the red grids using blue points Evaluate whether the blue points shown on the image accurately identify all targeted objects: red grids. Your assessment should be based on the following criteria: - Whether each blue point shown on the image accurately identify the targeted objects: red grids. - All targeted objects must be correctly identified (no targeted object is missed or left unhighlighted by blue points). - Each targeted object should correspond to exactly one blue point.
Answer 'true' if it is correct, otherwise answer 'false'.
Provide your evaluation strictly in the following JSON format:
{ "answer": true/false, "explanation": "If the answer is false, briefly summarize the main error."
}
**Answer:** False
**Explanation:** In the picture, the third grid in the first row, the fourth grid in the second row, and the third grid in the third row above the love heart are marked on both sides.

---

**State Value Evaluation** assesses whether a task has been successfully completed by analyzing the state of a game, robotics environment, or GUI as captured in an image. This evaluation includes:

- **Maze:** Evaluate model's ability to determinw whether the provided path in a maze constitutes a valid solution, without errors such as passing through walls.

---

**Illustrative Example of Maze Data**

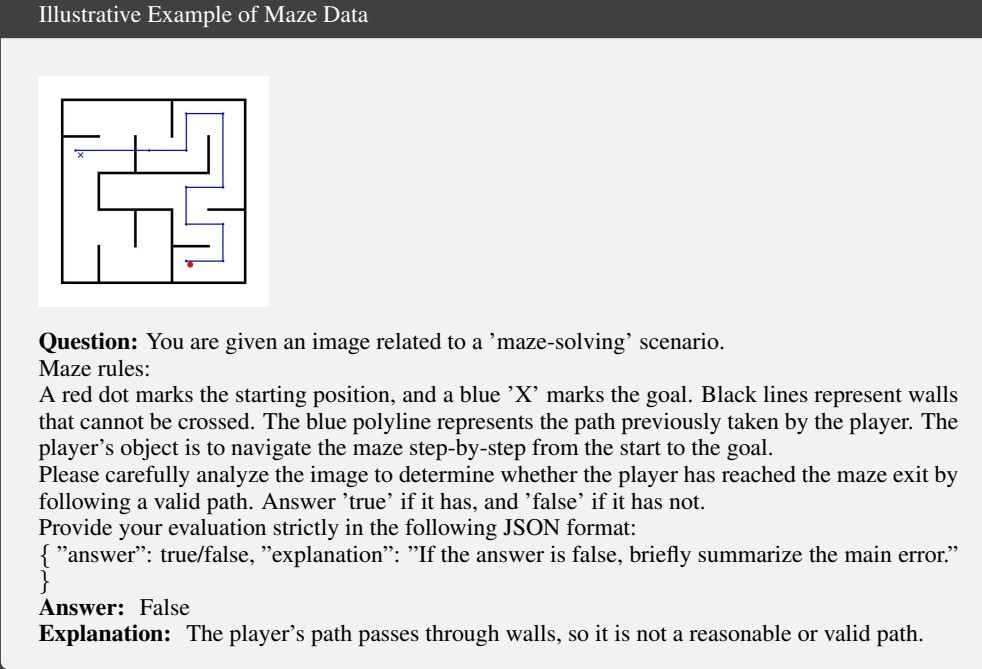

**Question:** You are given an image related to a 'maze-solving' scenario.
Maze rules:
A red dot marks the starting position, and a blue 'X' marks the goal. Black lines represent walls that cannot be crossed. The blue polyline represents the path previously taken by the player. The player's object is to navigate the maze step-by-step from the start to the goal.
Please carefully analyze the image to determine whether the player has reached the maze exit by following a valid path. Answer 'true' if it has, and 'false' if it has not.
Provide your evaluation strictly in the following JSON format:
{ "answer": true/false, "explanation": "If the answer is false, briefly summarize the main error."
}
**Answer:** False
**Explanation:** The player's path passes through walls, so it is not a reasonable or valid path.

---

- **FrozenLake:** Assesses both step-level and outcome-level correctness, determining whether the sequence of actions is reasonable and whether the target game state has been reached.

---

**Illustrative Example of FrozenLake Data**

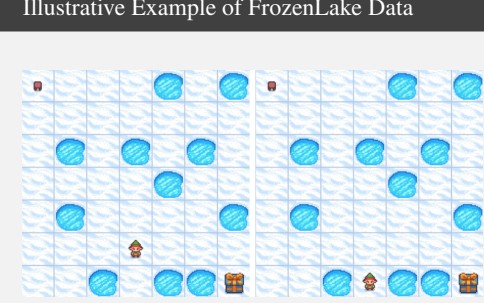

**Question:** You are given two images related to the 'FrozenLake' scenario.
FrozenLake rules:
- Player: The elf character.
- Exit: The treasure chest.
- Obstacles: The blue patches of water are holes and cannot be entered.
- Path: The white, snowy tiles are safe to walk on.
- Movement: The player can move one step at a time: up, down, left, or right
The two images show:
- The first image captures the player's current position at a given moment.
- The second image shows the player's position after making one move. In other words, these two images represent a single transition — before and after one move.
Please determine whether the player's move from the first image to the second image brings the game state closer to the goal state. Answer 'true' if the move brings the game state closer to the goal state, otherwise answer 'false'.
Provide your evaluation strictly in the following JSON format:
{ "answer": true/false, "explanation": "If the answer is false, briefly summarize the main error." }
**Answer:** False
**Explanation:** The player's action is not a reasonable step that leads toward the goal state.

---

- **Robotics:** Focuses on scenarios where a robot stacks blocks, requiring complex reasoning, and evaluates whether the model can accurately judge if the current block arrangement represents a valid sequence.

---

**Illustrative Example of Robotics Data**

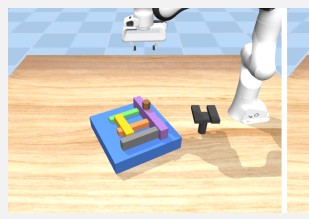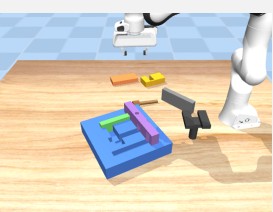

**Question:** This is a robotic arm stacking scenario where some blocks must be placed in a strict sequential order. You are given two images: the first shows the target configuration, and the second shows an intermediate state. Please analyze this intermediate state to determine if it follows the correct stacking sequence. Specifically, does this arrangement represent a valid and logical step towards reaching the target, without violating any placement order rules?
If the sequence is correct, please answer 'true'; otherwise, answer 'false'.
Provide your evaluation strictly in the following JSON format:
{ "answer": true/false, "explanation": "If the answer is false, briefly summarize the main error." }
**Answer:** False
**Explanation:** The orange block and the gray block are below the purple block, so the orange block and the gray block should be placed first. Therefore, the current state is incorrect.

- **GUI:** Covers mobile, PC, and web interfaces, evaluating whether the model can correctly determine if the bounding boxes in an image correspond to the buttons specified in the prompt.

---

**Illustrative Example of GUI Data**

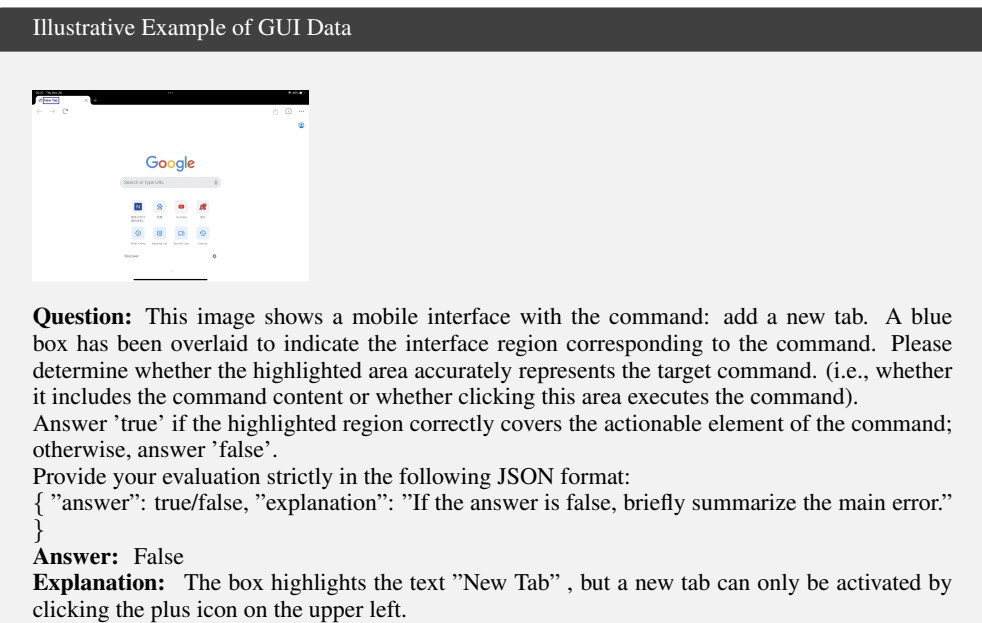

**Question:** This image shows a mobile interface with the command: add a new tab. A blue box has been overlaid to indicate the interface region corresponding to the command. Please determine whether the highlighted area accurately represents the target command. (i.e., whether it includes the command content or whether clicking this area executes the command).
Answer 'true' if the highlighted region correctly covers the actionable element of the command; otherwise, answer 'false'.
Provide your evaluation strictly in the following JSON format:
{ "answer": true/false, "explanation": "If the answer is false, briefly summarize the main error."
}
**Answer:** False
**Explanation:** The box highlights the text "New Tab" , but a new tab can only be activated by clicking the plus icon on the upper left.

---

**STEM** evaluates coding-related judgement tasks in STEM scenarios, assessing whether numerical values, variables, and other information in the image are consistent with the code. This evaluation includes:

- **Chart:** Evaluate whether, in coding tasks involving statistical visualizations (e.g., bar charts or histograms), the model can accurately determine if the rendered image fully matches the underlying code, including subtle details such as numerical values, colors, and legends.

---

**Illustrative Example of Chart Data**

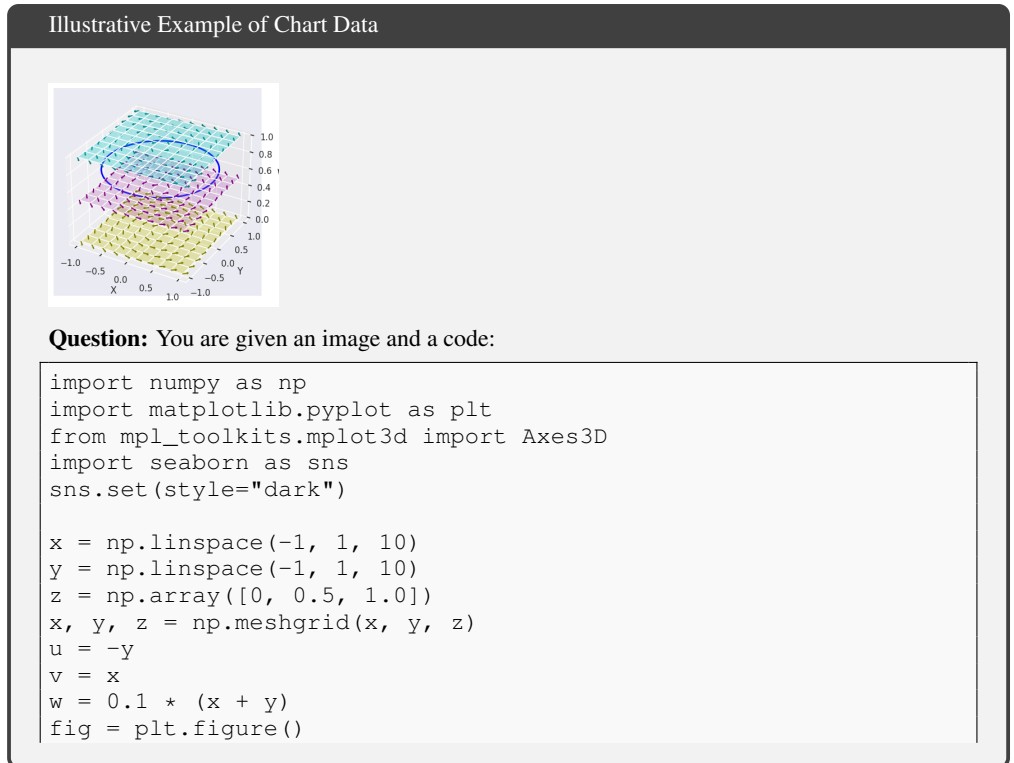

**Question:** You are given an image and a code:

```
import numpy as np
import matplotlib.pyplot as plt
from mpl_toolkits.mplot3d import Axes3D
import seaborn as sns
sns.set(style="dark")

x = np.linspace(-1, 1, 10)
y = np.linspace(-1, 1, 10)
z = np.array([0, 0.5, 1.0])
x, y, z = np.meshgrid(x, y, z)
u = -y
v = x
w = 0.1 * (x + y)
fig = plt.figure()
```

```python
ax = fig.add_subplot(111, projection='3d')
ax.quiver(x, y, z, u, v, w, length=0.1, normalize=True, color=['
    olive', 'purple', 'teal'])
ax.plot_surface(x[:, :, 0], y[:, :, 0], z[:, :, 0], color='
    yellow', alpha=0.3)
ax.plot_surface(x[:, :, 1], y[:, :, 1], z[:, :, 1], color='
    violet', alpha=0.3)
ax.plot_surface(x[:, :, 2], y[:, :, 2], z[:, :, 2], color='cyan
    ', alpha=0.3)

theta = np.linspace(0, 2 * np.pi, 100)
x_circle = np.cos(theta)
y_circle = np.sin(theta)
z_circle = np.ones_like(theta) * 0.25
ax.plot(x_circle, y_circle, z_circle, color='blue', linewidth=2)

ax.set_xlabel('X')
ax.set_ylabel('Y')
ax.set_zlabel('W')
plt.show()
```

Please carefully analyze whether the image was generated using this code. You need to closely examine whether the image and the code match exactly, including the legend, variable names, line color, line style, plotting range, data values, and other details. Answer 'true' if it does, and 'false' if it doesn't.

Provide your evaluation strictly in the following JSON format:
{ "answer": true/false, "explanation": "If the answer is false, briefly summarize the main error." }

**Answer:** False

**Explanation:** The code shows the z-coordinate of the blue circle is set to 0.25, whereas the image shows the blue circle is located at a z-coordinate of 0.75, which is inconsistent, so the code does not match the image.

- **LaTeX:** Assess whether, given LaTeX code and its corresponding image, the model can precisely verify a complete correspondence, capturing fine-grained elements such as values, variables, and operators.

---

**Illustrative Example of LaTeX Data**

$$\mathcal{H}^{(0)} = -\frac{2\kappa}{\sqrt{-\gamma}}\frac{\delta I^{(0)}}{\delta\gamma_{\mu\nu}}\frac{\delta I^{(0)}}{\delta\gamma_{\lambda\rho}}\left(\gamma_{\mu\rho}\gamma_{\nu\lambda} - \frac{1}{d-1}\gamma_{\mu\nu}\gamma_{\lambda\rho}\right) + \frac{\sqrt{-\gamma}}{\kappa}\Lambda = 0,$$

**Question:** Please carefully analyze this image and determine whether it was generated by compiling the given LaTeX code:

```latex
\begin{align*}
{\cal H}^{(0)} = -\frac{2 \kappa}{\sqrt{-\gamma}}
\frac{\delta I^{(0)}}{\delta \gamma_{\mu \nu}}
\frac{\delta I^{(0)}}{\delta \gamma_{\lambda \rho}}
\left( \gamma_{\mu \rho} \gamma_{\nu \lambda}
- \frac{1}{d-2} \gamma_{\mu \nu } \gamma_{\lambda \rho} \right)
+ \frac{\sqrt{-\gamma}}{\kappa} \Lambda = 0,
\end{align*}
```

Answer 'true' if it does, and 'false' if it doesn't.
Provide your evaluation strictly in the following JSON format:
{ "answer": true/false, "explanation": "If the answer is false, briefly summarize the main error." }

**Answer:** False

**Explanation:** In the LaTeX code, the denominator of the fraction inside the parentheses is d-2, whereas in the image, the corresponding denominator is d-1, which presents a contradiction, so the code does not match the image.

## A.2 BENCHMARK CURATION PIPELINE

ViVerBench is constructed through a systematic pipeline following two important criteria: **(1) ensuring sufficient difficulty** and **(2) guaranteeing answer correctness with no room for dispute**. The data construction process consists of four stages:

**Initial Dataset Construction**   We employ three strategies to construct data for the 16 tasks:

- **Manual Annotation** For tasks in {Object, Attribute, Non-spatial, Static Physics, Counting}, the lack of high-quality verification datasets and the difficulty of constructing error-free, unambiguous data necessitate expert involvement. We invited 12 domain experts to curate challenging datasets. Starting from perfectly matched image–prompt pairs, we applied fine-grained editing, inpainting, and prompt modification to create false examples, each accompanied by detailed error explanations. Considerable effort was devoted to ensure both diversity and difficulty throughout data construction and selection.
- **Programmatic Data Generation** For tasks in {Spatial, Maze, Frozenlake, Robotics}, we developed tailored scripts to systematically generate true/false examples, along with their corresponding explanations.
- **Augmented Open-source Data** For tasks in {Abstract Patterns, Dynamic Physics, Bounding Box, Pointing, GUI, Charts, LaTeX}, we collected high-quality samples from open-source datasets (Feng et al., 2025; Bordes et al., 2025; Cheng et al., 2025; Wu et al., 2024; Hao et al., 2025) and further created challenging true and false examples with detailed explanations, specifically designed for visual verification tasks.

**Expert Review and Difficulty Enhancement**   The initially constructed data were reviewed with a thorough review by five new experts, who verified whether the true examples were strictly correct and free of ambiguities and whether the explanations of the false example were reasonable and correct. They also assessed task difficulty, providing higher-difficulty annotations for tasks such as Object, Abstract Patterns, and Bounding Box.

**Human Evaluation**   To further validate the dataset, we recruited ten experts for human evaluation. Since many true/false examples are paired from the same prompt or image, the data were evenly split into two groups to prevent prior bias, ensuring that paired samples were placed in different groups. Each group was independently evaluated by five experts.

**Dataset Refinement and Final Selection**   Based on human evaluation results, we identified questions that were frequently answered incorrectly and subjected them to an additional correctness check by five new experts. Incorrect items were removed, and questions with potential ambiguities were refined. This process yielded **ViVerBench**, a comprehensive and challenging visual verification benchmark with 3,594 data samples.

## A.3 EVALUATION METHODS

We evaluate model performance using two complementary metrics: rule-based evaluation and model-based evaluation. Let $N$ be the total number of one task in **ViVerBench**. For each sample $i$, we denote the ground-truth answer as $y_i \in \{\text{true}, \text{false}\}$ and the model-predicted answer as $\hat{y}_i \in \{\text{true}, \text{false}\}$. When $y_i = \text{false}$, the benchmark provides a ground-truth explanation $e_i$, and when $\hat{y}_i = \text{false}$, the model is required to generate an explanation $\hat{e}_i$. To assess explanation quality, we use a judge model (such as GPT-4.1), denoted as $\mathcal{F}(e_i, \hat{e}_i)$, which returns true if $e_i$ and $\hat{e}_i$ are considered consistent. We further denote by $\mathbf{1}(\cdot)$ the indicator function that outputs 1 if the condition holds and 0 otherwise.

**Rule-based Evaluation**   Rule-based evaluation measures only the correctness of the predicted answer. The accuracy is computed as:

$$\text{Acc}_{\text{rule-based}} = \frac{1}{N} \sum_{i=1}^{N} \mathbf{1}(\hat{y}_i = y_i).$$ (1)

This metric ignores explanations and focuses solely on answer correctness.

**Model-based Evaluation** Model-based evaluation extends the rule-based setting by additionally requiring explanation consistency when both the ground truth and the prediction are false. The accuracy is defined as:

$$\text{Acc}_{\text{model-based}} = \frac{1}{N} \left[ \sum_{i:y_i=\text{true}} \mathbf{1}(\hat{y}_i = y_i) + \sum_{i:y_i=\text{false}} \mathbf{1}(\hat{y}_i = y_i) \cdot \mathbf{1}(\mathcal{F}(e_i, \hat{e}_i)) \right]. \qquad (2)$$

When $y_i$ is true, a prediction is correct only if $\hat{y}_i$ matches $y_i$. When $y_i$ is false, correctness requires not only $\hat{y}_i = y_i$ but also judge model verifies consistency between $\hat{e}_i$ and $e_i$. This stricter metric prevents spurious correctness from random guessing by enforcing explanation validity in false cases.

## B EVALUATION ON GENERAL BENCHMARKS

**GenRM Evaluation** For the evaluation of Vision-Language Generative Reward Models, we use VL-RewardBench (Li et al., 2025), focusing on the text-outcome verification capability of GenRM. As shown in Table 4, we observe that OmniVerifier-7B achieves substantial and comprehensive improvements over the baseline, particularly in reducing hallucination. This shows that visual outcome verification can effectively generalize to text-outcome verification, which we attribute to the shared atomic capabilities underlying cross-modal verification.

Table 4: Evaluation Results on VLRewardBench.

| Model | General | Hallucination | Reasoning | Overall Accuracy | Macro Avg Accuracy |
|-------|---------|---------------|-----------|------------------|--------------------|
| Qwen 2.5-VL 7B | 37.16 | 45.79 | 54.40 | 46.72 | 45.79 |
| OmniVerifier 7B | 41.53 (**+4.37**) | 70.09 (**+24.3**) | 57.86 (**+3.46**) | 62.80 (**+16.08**) | 56.49 (**+10.7**) |

**Evaluation on Mainstream Benchmarks** To examine more general and more complex tasks, we additionally evaluate the models on eight mainstream perception and image-reasoning benchmarks (Chen et al., 2024; Tong et al., 2024; Wang et al., 2024; Hao et al., 2025; Xu et al., 2025; Roberts et al., 2025; Liu et al., 2023b).

Table 5: Evaluation Results on Mainstream Perception and Image-reasoning Benchmarks.

| Model | MMStar | MMVP | RealWorldQA | MathVision | EMMA | VisuLogic | ZeroBench | OCRBench |
|-------|--------|------|-------------|------------|------|-----------|-----------|----------|
| Qwen 2.5-VL 7B | 61.7 | 72.9 | 68.8 | 22.1 | 24.8 | 26.9 | 13.7 | 85.1 |
| OmniVerifier 7B | 63.9 (+2.2) | 77.7 (+4.8) | 68.1 (-0.7) | 25.2 (+3.1) | 29.4 (+4.6) | 25.4 (-1.5) | 14.4 (+0.7) | 87.1 (+2.0) |

As shown in Table 5, OmniVerifier demonstrates clear improvements on most benchmarks. This suggests that a critic model trained with RL on pointwise samples can generalize into a strong policy model, leading to broad gains across diverse downstream generation tasks. This finding aligns with observations from LLaVA-Critic-R1 (Wang et al., 2025b), and we further validate it in the visual-outcome setting. We argue that this observation, namely critics enhance generation, highlights a highly promising direction for future large-scale model development: a strong policy model can be trained into a strong verifier or critic model, and the two can mutually reinforce one another, enabling a multimodal system capable of self-evaluation and ultimately self-improvement. This also highlights the importance of a universal verifier.

## C ANALYSIS OF TEST-TIME SCALING EFFICIENCY

We report the average number of iterative refinement rounds when using Qwen-Image and GPT-Image-1 on T2I-ReasonBench and GenEval++ benchmarks:

From the Table 6, we can observe the following:

- **Sequential TTS outperforms Parallel TTS in both speed and performance.** On GenEval++, Sequential TTS requires fewer than two refinement rounds on average to achieve higher-quality results. In contrast, Parallel TTS generates 10 images per prompt without demonstrating significant performance gains.

Table 6: Performance and Refinement Efficiency Across Backbones, Verifiers, and TTS Strategies.

| Backbone | Verifier | Test-Time Scaling | T2I-ReasonBench | | GenEval++ | |
|---|---|---|---|---|---|---|
| | | | Overall | Mean Round | Overall | Mean Round |
| **Qwen-Image** | - | - | 55.5 | 1 | 0.675 | 1 |
| **GPT-Image-1** | - | - | 76.8 | 1 | 0.689 | 1 |
| **Qwen-Image** | OmniVerifier-7B | Parallel | 58.1 | 10 | 0.693 | 10 |
| **GPT-Image-1** | OmniVerifier-7B | Parallel | 78.1 | 10 | 0.700 | 10 |
| **Qwen-Image** | Qwen 2.5-VL 7B | Sequential | 57.4 | 4.67 | 0.682 | 1.95 |
| **Qwen-Image** | OmniVerifier-7B | Sequential | 59.2 | 3.86 | 0.718 | 1.86 |
| **GPT-Image-1** | Qwen 2.5-VL 7B | Sequential | 77.8 | 2.01 | 0.693 | 1.33 |
| **GPT-Image-1** | OmniVerifier-7B | Sequential | 79.3 | 1.59 | 0.721 | 1.27 |

- **OmniVerifier substantially reduces hallucinations and improves judgment accuracy.** Across all backbones, employing OmniVerifier consistently requires fewer refinement rounds than using Qwen 2.5-VL. RL training equips OmniVerifier with stronger visual-outcome verification capabilities, which mitigates hallucinations and produces accurate explanations and edit prompts. Consequently, higher-quality results are attained with fewer refinement rounds.

- **Stronger backbones reduce verification rounds, shortening inference time.** GPT-Image-1 demonstrates clear advantages over Qwen-Image in reasoning and world knowledge. On T2I-ReasonBench, this translates to nearly threefold faster inference. Stronger backbones also follow instructions more accurately, enabling precise edits and refinements. We therefore anticipate that, as more powerful unified models emerge, OmniVerifier-TTS will consistently deliver faster inference and higher-quality results.

## D  RAISING THE PERFORMANCE CEILING WITH STRONG VERIFIERS

We have added the results of using Gemini 2.5 Pro as the visual verifier on T2I-ReasonBench and GenEval++, as shown in Table 7:

Table 7: Results of Sequential TTS with Gemini 2.5 Pro on T2I-ReasonBench and GenEval++.

| Backbone | Verifier | Test-Time Scaling | T2I-ReasonBench Overall | GenEval++ Overall |
|---|---|---|---|---|
| **Qwen-Image** | - | - | 55.5 | 0.675 |
| **Qwen-Image** | Omniverifier-7B | Sequential | 59.2 | 0.718 |
| **Qwen-Image** | Gemini 2.5 Pro | Sequential | 62.7 | 0.736 |
| **GPT-Image-1** | - | - | 76.8 | 0.689 |
| **GPT-Image-1** | Omniverifier-7B | Sequential | 79.3 | 0.721 |
| **GPT-Image-1** | Gemini 2.5 Pro | Sequential | 81.6 | 0.746 |

Due to Gemini-2.5-Pro's strong world knowledge and visual-judgment abilities, it achieves significantly better results on both benchmarks. We also find that incorporating TTS with a strong verifier pushes the baseline into a substantially higher performance tier, markedly extending its achievable upper bound. Looking ahead, we plan to further improve the model by scaling up both its size and the amount of high-quality training data.

## E  ADDITIONAL VISUALIZATION RESULTS FOR OMNIVERIFIER-TTS

In this section, we provide a detailed analysis of the strengths and weaknesses of Sequential TTS versus Parallel TTS.

Sequential TTS significantly raises the generation performance ceiling of unified model. Through multiple rounds of self-refinement, OmniVerifier continuously leverages its world knowledge, reasoning ability, and critic capability to guide visual generation through fine-grained editing. This allows it to handle highly complex compositional prompts, including domains unseen in the training data, as shown in Fig. 8. In contrast, for such highly challenging prompts, Parallel TTS struggles to generate accurate outputs regardless of the number of samples. This is because it does not perform targeted reflection and correction for the global prompt condition; its trial-and-error approach without self-reflection makes it difficult to achieve breakthroughs in complex scenarios.

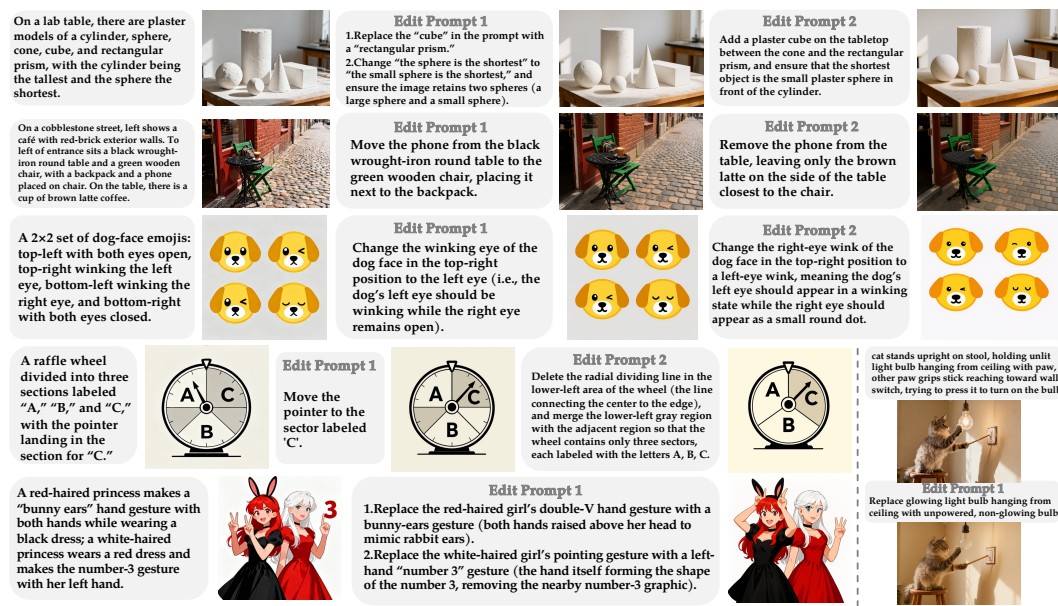

Figure 8: More results of OmniVerifier-TTS.

However, Sequential TTS is constrained by the backbone's editing capability and is subject to error accumulation, which affects the final image quality. In Fig. 9, we present some failure cases of Sequential TTS using GPT-Image-1. While the final image aligns with the prompt, GPT-Image-1 exhibits limited robustness to its own generated image distribution, causing a gradual yellowing of images over successive edits. Although OmniVerifier generally provides correct edit prompts, a weak backbone editing capability prevents accurate application of these changes. Consequently, errors accumulate across multiple rounds of self-refinement, ultimately degrading the final image quality. This highlights that mitigating error accumulation is a critical challenge for future unified models tasked with multi-round generation.

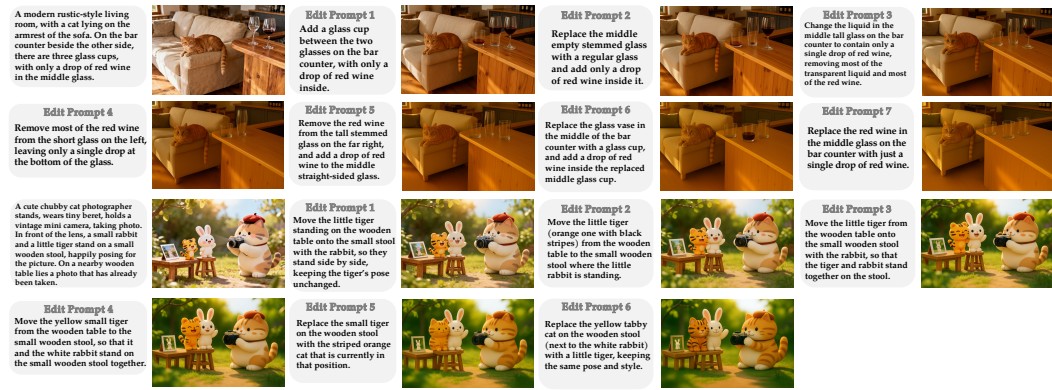

Figure 9: Failure cases in image quality caused by the backbone's limited editing capability.

## F    LONGCOT PATTERN ANALYSIS OF OMNIVERIFIER

We train OmniVerifier with end-to-end reinforcement learning, without requiring any cold start. The training objectives include a format reward to regulate reasoning behavior and a rule-based reward to assess answer correctness. After training, we observe that the model autonomously learns a high-quality and well-structured chain-of-thought (CoT) pattern. This humanlike CoT reasoning is precisely what visual verifiers need in complex scenarios, as illustrated below:

---

### Illustrative Examples of OmniVerifier LongCoT Patterns (1)

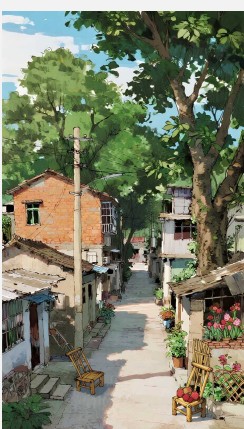

**Question:** This image was generated from the prompt: "Above the picture are the blue sky and white clouds, and there are trees on both sides and in the distance. There are residential houses on both sides of the alley. On the left side, from near to far, there are low bungalows and a two-story red-brick building. There is a red lantern hanging under the eaves of the small cottage in the middle of the left side of the picture. There are flower pots placed beside the small cottage closest to the camera on the right side. There is an empty chair on the left side of the middle alley, and red fruits are placed on the chair on the right side. A utility pole stands on the left of the scene."

Please carefully analyze the image and determine whether all the objects and their quantities mentioned in the prompt are correctly represented in the image. If all the objects and quantities are correctly presented, please answer 'true'; otherwise, answer 'false'. Provide your evaluation strictly in the following JSON format:

{ "answer": true/false, "explanation": "If the answer is false, briefly summarize the main error." }

---

**OmniVerifier-7B:**

```
<think>
Let's break down the prompt and compare it to the image:
1.  **Blue sky and white clouds**:  The image shows a blue sky with white clouds, which
is correctly represented.
2.  **Trees on both sides and in the distance**:  There are trees on both sides of the
alley and in the distance, which is correctly represented.
3.  **Residential houses on both sides of the alley**:  There are residential houses on
both sides of the alley, which is correctly represented.
4.  **Low bungalows and a two-story red-brick building on the left side**:  The left
side of the image shows low bungalows and a two-story red-brick building, which is
correctly represented.
5.  **Red lantern hanging under the eaves**:  There is no red lantern visible in the
image, so this is not correctly represented.
6.  **Flower pots beside the small cottage closest to the camera on the right side**:
There are flower pots visible beside the small cottage on the right side, which is
correctly represented.
7.  **Empty chair on the left side of the middle alley**:  There is an empty chair on
the left side of the middle alley, which is correctly represented.
8.  **Red fruits on the chair on the right side**:  There are red fruits on the chair
on the right side, which is correctly represented.
9.  **Utility pole on the left of the scene**:  There is a utility pole on the left of
the scene, which is correctly represented.
The main error in the prompt is the absence of a red lantern, which is not present in
the image.  However, the rest of the objects and quantities mentioned are correctly
represented in the image.
</think>

{ "answer":  false, "explanation":  "The main error is the absence of a red lantern
hanging under the eaves of the small cottage in the middle of the left side of the
picture."}
```

---

**Illustrative Examples of OmniVerifier LongCoT Patterns (2)**

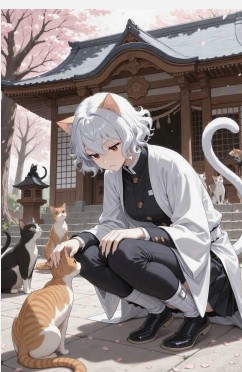

**Question:** This image was generated from the prompt: "A cartoon girl with a cat's tail, short curly white hair, and sharp nails is smiling with her lips closed. She is wearing a Demon Slayer Corps uniform and a white long - sleeved haori jacket. She is squatting down, looking at a little cat and gently stroking it. There are also many kittens surrounding her. The background features East Asian - style shrine buildings and cherry blossom trees. "
Please carefully analyze the image and determine whether all the attributes specified in the prompt (such as color, texture, shape, material, lighting, expression, and motion) are correctly represented in the image. If all the attributes are correctly presented, please answer 'true'; otherwise, answer 'false'. Provide your evaluation strictly in the following JSON format:
{ "answer": true/false, "explanation": "If the answer is false, briefly summarize the main error." }

---

**OmniVerifier-7B:**

```
<think>
Let's break down the attributes mentioned in the prompt and compare them to the image:
1.  **Cartoon girl with a cat's tail**:  The character in the image has a cat's tail.
2.  **Short curly white hair**:  The character has short, curly white hair.
3.  **Sharp nails**:  The character's nails are not explicitly mentioned as sharp, but
they appear to be normal.
4.  **Smiling with her lips closed**:  The character is not smiling; she appears to be
frowning.
5.  **Demon Slayer Corps uniform**:  The character is wearing a uniform that resembles
the one from the Demon Slayer anime, but it's not explicitly labeled as such.
6.  **White long-sleeved haori jacket**:  The character is wearing a white long-sleeved
haori jacket.
7.  **Squatting down**:  The character is squatting down.
8.  **Looking at a little cat**:  The character is looking at a cat.
9.  **Gently stroking it**:  The character is gently stroking a cat.
10.  **Many kittens surrounding her**:  There are multiple cats surrounding the
character.
11.  **East Asian-style shrine buildings**:  The background features East Asian-style
shrine buildings.
12.  **Cherry blossom trees**:  The background features cherry blossom trees.
The character is not smiling, which contradicts the prompt.  However, the rest of the
attributes are correctly represented in the image.
</think>

{ "answer":  false, "explanation":  "The main error is that the character is not
smiling, which contradicts the prompt's description."}
```

---

We find that by decomposing a complex problem into a series of specific and simpler subproblems, this structured and organized reasoning style can greatly improve the model's verification accuracy. Moreover, as discussed in Section 5, we observe that training with rule-based rewards without direct supervision on textual outputs does not significantly affect text modeling. The model is still capable of generating coherent LongCoT reasoning, explanations, and edit prompts for TTS self-refinement.

---

**Finding 5.** *Minimal Supervision Suffices for Generalization*
*Optimizing only for binary true/false outcomes via rule-based reinforcement learning enhances verification while preserving the model's explanatory language capacity. Thus, task-specific supervision over explanations is not required to maintain language modeling quality.*

---

## G   BROADER WORLD-MODELING INTERLEAVED REASONING

We further employ task-specific data to train verifiers for maze navigation and robotics, in order to explore whether OmniVerifier can provide meaningful critiques in broader world modeling reasoning scenarios. As shown in Fig. 10, we use Qwen2.5-VL-72B (Bai et al., 2025) as the policy model. In the maze scenario, where each action corresponds to moving up, down, left, or right, the policy model often makes mistakes such as walking through walls. With the strong visual-outcome reflection capability of OmniVerifier, these errors can be promptly corrected. Similarly, in the robotics scenario, where the placement of blocks follows a logical order, OmniVerifier provides powerful error correction for the policy model, substantially improving accuracy in such reasoning tasks.

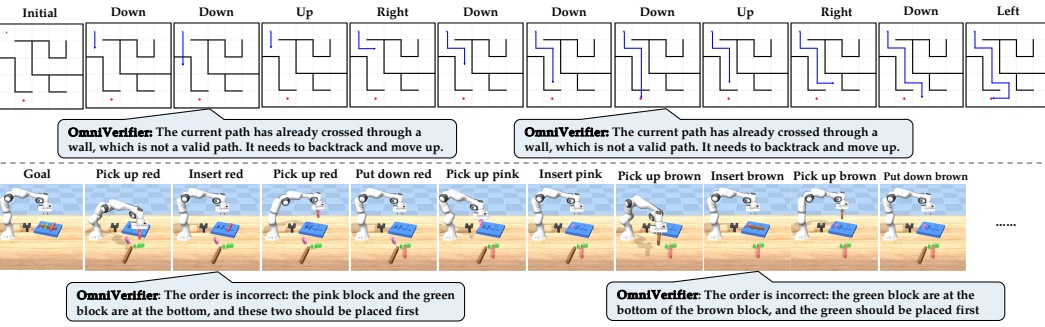

Figure 10: Extend OmniVerifier to world-modeling interleaved reasoning tasks: Maze and Robotics.

## H   PROMPTS USED IN THE TWO AUTOMATED PIPELINES FOR VISUAL VERIFIER DATA CONSTRUCTION

---

**Method1: Prompt for Image Recaption**

```
You are a powerful Image Captioner.
Given an image, you must generate a caption that is **accurate and faithful** to the
visual content.
You must strictly follow these guidelines:
1.  Accuracy First
• Describe only the elements that can be clearly identified in the image (objects,
attributes, spatial relationships, scenes, etc.).
• No guesses, subjective assumptions, or irrelevant details are allowed.
• Do not fabricate elements that are not present in the image.
2.  Description Elements
• Focus on object attributes (color, texture, shape, material).
• Include visible scene elements (background, environment, lighting).
• Cover actions, poses, expressions, and spatial relationships.
• Accurately capture any numbers, letters, or quantities that are discernible.
3.  Language Style
• Use an objective and neutral tone.
• Avoid subjective evaluations (e.g., \beautiful," \cute").
• Keep the caption clear and precise, without unnecessary embellishment.
• The length of the caption should be determined by the complexity of the image, but it
must not exceed 60 words.
Output Format Requirement:  Output only one caption, without any explanation or
additional text.
```

---

**Method1: Prompt for Caption Modification**

```
You are a powerful image caption editor.
Your task is to make **subtle yet important modifications** to key attribute details
in an image caption (e.g., color, texture, shape, material, pose, facial expression,
numbers, letters, quantities, etc.).  The requirement is that the modified detail must
be **significant enough that an image generated from the original caption would not
satisfy the modified caption**.
**Rules:**
1.  You are allowed to make **only one modification**.
2.  Your modifications should be **diverse** and not limited to a single type of
attribute(such as color).  Any attribute-related description can be the target of
```

```
your edits, such as shape, number, color, letter, quantity, expression, pose, and so
on.
3. Vague or ambiguous modifications (e.g., changing \white" to \off-white," or \some"
to \a few") are **not valid** because their boundaries are hard to define.
4. You must output the json format, with the following two keys:
• The modified prompt (after your edit).
• The specific key detail you changed.
**Example 1:**
**Input:**
A movie scene: a black-and-white photograph showing an old man sitting inside a car.
He is wearing a mask and looking into the camera through the front rearview mirror. A
bird-shaped ornament hangs on the left side of the rearview mirror. There is also a
GPS navigation device below the center console, with the screen displaying the number
'700'. Other parts of the car include some buttons and controls, creating a simple and
cozy atmosphere. The background is blurred, with a faint view of the outside.
**Output:**
{
    Modified prompt: "A movie scene: a black-and-white photograph showing an old man
sitting inside a car. He is wearing a mask and looking into the camera through the
front rearview mirror. A bird-shaped ornament hangs on the left side of the rearview
mirror. There is also a GPS navigation device below the center console, with the
screen displaying the number '710'. Other parts of the car include some buttons and
controls, creating a simple and cozy atmosphere. The background is blurred, with a
faint view of the outside.",
    Changed detail: "The screen number was changed from '700' to '710'."
}
**Now your turn:**
**Input:**
**Output:**
```

## Method2: Prompt for Image Recaption

```
**Image Captioning Instructions **
**Role** You are a high-precision image captioner.
Your goal is to generate an accurate, concise, and fact-based caption based on the
given image with a red bounding box indicating the target object: object name.
**Generation Rules**
1. **Object Inclusion**
• The caption **must mention the object inside the bounding box**, describing its
appearance and location accurately.
• The caption should also include other important objects in the scene.
2. **Attribute Details**
• Describe observable attributes of objects, such as color, texture, shape, material,
lighting, facial expressions, and actions.
3. **Spatial Relationships**
• Clearly describe spatial relations between objects (e.g., top/bottom, left/right,
front/back, near/far).
4. **Interactions**
• Describe actions or interactions between objects (e.g., \sitting on," \holding,"
\leaning against").
5. **Factual Accuracy**
• Only describe content that can be confirmed from the image.
• Do not add guesses, subjective imagination, or irrelevant details.
• If an attribute or detail cannot be confirmed, do not mention it.
6. **Distinguishing Similar Objects**
• If there are other objects of the same category in the image, clearly distinguish and
describe the bounding box object's specific location and features.
7. **Explantion**
In addition to the caption, you need to provide an explanation indicating where the
generated image is incorrect|assuming that all parts of the image outside the bbox are
correctly generated, specify the error within the bbox (i.e., the object inside the
bbox was not generated correctly).
8. **Length Limit**
• The caption must not exceed 60 English words.
**Loop-Enabled Self-Correction Process**
**Step 1 | Initial Draft**
• Generate a caption following all rules.
**Step 2 | Self-Check**
• Verify the following:
1. Caption explicitly mentions the bounding box object.
2. Object's attributes and location are accurately described.
3. Spatial and interaction relationships are included when visible.
4. Description is based only on observable facts (no guesses or imaginary details).
5. Explanation is reasonable.
6. Caption length is < 60 words.
**Step 3 | Correction Loop**
• If **any** check fails:
```

```
• Revise the caption to meet all rules.
• Re-run the self-check.
• Repeat until all checks pass.
**Final Output Format**
Always output only the **final, corrected** caption and explanation in this format:
{    caption:     explanation:  }
**Example**
**Input** Bounding box object:  teddy bear
**Output** {    caption:  "Two girls stand on grass.  The girl on the left, wearing
a dress, holds a teddy bear.  The girl on the right holds a red balloon, with a white
house visible in the distance.",    explanation:  "The generated image is missing a
teddy bear." }
**Input** Bounding box object:
**Output**
```

## I    LIMITATIONS AND FUTURE WORKS

Although OmniVerifier and OmniVerifier-TTS demonstrate promising performance, we identify two key limitations that suggest directions for future work:

One limitation is that certain tasks may generalize less effectively. For tasks such as maze involve a large domain gap, and optimizing for them requires task-specific data. We posit that a truly universal verifier should perform robustly across diverse tasks. Future work will explore strategies in training and data construction to enhance OmniVerifier's generalization, moving closer to a genuinely universal verifier.

OmniVerifier-TTS is influenced by the backbone. Currently, due to issues with training data and strategies, unified multimoda model is sensitive to the distribution of the images it generates or edits. Some models exhibit unusual behaviors under multi-step self-refinement; for instance, GPT-Image-1 tends to produce increasingly yellowish images after iterative edits. Importantly, these artifacts affect only style and do not compromise verification performance. We view this as a subtle limitation of the backbone rather than of OmniVerifier-TTS itself, and we encourage further efforts to enhance style consistency under multi-step self-refinement.

## J    USE OF LARGE LANGUAGE MODELS

We used a large language model solely to assist in polishing English writing and improving clarity. All research ideas, experiments, results, and interpretations are entirely our own.

