# OpenReview forum: "Generative Universal Verifier as Multimodal Meta-Reasoner"
_ICLR.cc/2026/Conference — ICLR 2026 Oral_

### Official Review · Reviewer_Q1VF · 2025-10-19

**Soundness:** 3
**Presentation:** 3
**Contribution:** 4
**Rating:** 8
**Confidence:** 5

**Summary:**

This paper introduces Generative Universal Verifier, a new approach to help AI models better verify and improve visual outputs during reasoning. The authors first create ViVerBench, a comprehensive benchmark showing current models struggle with visual verification tasks. They then develop automated methods to build training data and train OmniVerifier-7B, which significantly improves verification performance. Finally, they apply this verifier through OmniVerifier-TTS, a sequential refinement system that enhances image generation quality through iterative verification and editing.

**Strengths:**

1. The structure of this paper is clear, which investigates three central questions.
2. The ViVerBench benchmark is well-designed with 16 diverse task categories and careful human validation, providing a solid foundation for evaluating visual verification capabilities.
3. The identification of three atomic capabilities (Explicit Alignment, Relational Verification, Integrative Reasoning) offers important insights into how visual verification works and generalizes.
4. The sequential TTS approach shows clear advantages over parallel methods like Best-of-N, achieving better results with fewer generation steps.

**Weaknesses:**

1. Computational costs of running multiple verification and editing steps aren't adequately addressed, which could limit practical application. The authors should discuss the extra inference cost when adopting OmniVerifier-TTS.
2. As shown in Table 1, the core component, OmniVerifier-7B, achieves an overall score of only 0.653 on ViVerBench. Relying on such a judge can introduce a fundamental risk into the entire iterative refinement process.
3. While the authorsdemonstrate in Table 2 that their specifically trained OmniVerifier-7B judge outperforms QwenVL, they do not explore whether a judge with superior foundational abilities would yield further gains. For instance, as shown in Table 1, Gemini-2.5-Pro achieves the highest score on the ViVerBench benchmark. Would a more powerful judge like Gemini-2.5-Pro lead to significantly better TTS results？

**Questions:**

1. What are the specific failure cases where sequential TTS performs worse than parallel approaches?

---

> ### Author Response · Authors · 2025-11-20
> **Response to Reviewer Q1VF (Part 1/2)**
>
> We thank you for your thorough review. We appreciate the positive comments, as well as the thoughtful points raised. Please kindly find our response to your comments below. Additionally, all modifications to the manuscript have been highlighted in blue for easy reference. Please feel free to let us know if you have any additional concerns or questions.
> >**Q1: Computational costs of running multiple verification and editing steps aren't adequately addressed, which could limit practical application. The authors should discuss the extra inference cost when adopting OmniVerifier-TTS.**
>
> **A1:** Thank you for the suggestion. We report the average number of iterative refinement rounds when using Qwen-Image and GPT-Image-1 on T2I-ReasonBench and GenEval++ benchmarks:
>
> **Table 1: Performance and Refinement Efficiency Across Backbones, Verifiers, and TTS Strategies.**
> | Backbone    | Verifer         | Test-Time Scaling | T2I-ReasonBench Overall | T2I-ReasonBench Mean Round | GenEval++ Overall | GenEval++ Mean Round |
> | ----------- | --------------- | ----------------- | ----------------------- | -------------------------- | ----------------- | -------------------- |
> | Qwen-Image  | -               | -                 | 55.5                    | 1                          | 0.675             | 1                    |
> | GPT-Image-1 | -               | -                 | 76.8                    | 1                          | 0.689             | 1                    |
> | Qwen-Image  | Omniverifier-7B | Parallel          | 58.1                    | 10                         | 0.693             | 10                   |
> | GPT-Image-1 | Omniverifier-7B | Parallel          | 78.1                    | 10                         | 0.700             | 10                   |
> | Qwen-Image  | Qwen2.5-VL-7B   | Sequential        | 57.4                    | 4.67                       | 0.682             | 1.95                 |
> | Qwen-Image  | Omniverifier-7B | Sequential        | 59.2                    | 3.86                       | 0.718             | 1.86                 |
> | GPT-Image-1 | Qwen2.5-VL-7B   | Sequential        | 77.8                    | 2.01                       | 0.693             | 1.33                 |
> | GPT-Image-1 | Omniverifier-7B | Sequential        | 79.3                    | 1.59                       | 0.721             | 1.27                 |
>
> From the table, we can observe the following:
>
> 1.	**Sequential TTS outperforms Parallel TTS in both speed and performance.** On GenEval++, Sequential TTS requires fewer than two refinement rounds on average to achieve higher-quality results. In contrast, Parallel TTS generates 10 images per prompt without demonstrating significant performance gains.
>
> 2.	**OmniVerifier substantially reduces hallucinations and improves judgment accuracy.** Across all backbones, employing OmniVerifier consistently requires fewer refinement rounds than using Qwen 2.5-VL. RL training equips OmniVerifier with stronger visual-outcome verification capabilities, which mitigates hallucinations and produces more accurate explanations and edit prompts. Consequently, higher-quality results are attained with fewer refinement rounds.
>
> 3.	**Stronger backbones reduce verification rounds, shortening inference time.** GPT-Image-1 demonstrates clear advantages over Qwen-Image in reasoning and world knowledge. On T2I-ReasonBench, this translates to nearly threefold faster inference. Stronger backbones also follow instructions more accurately, enabling precise edits and refinements. We therefore anticipate that, as more powerful unified models emerge, OmniVerifier-TTS will consistently deliver faster inference and higher-quality results.
>
> We have added this experiment in **Appendix C of the updated manuscript**. Thank you again for your valuable feedback.
>
> >**Q2: As shown in Table 1, the core component, OmniVerifier-7B, achieves an overall score of only 0.653 on ViVerBench. Relying on such a judge can introduce a fundamental risk into the entire iterative refinement process.**
>
> **A2:** It is worth noting that OmniVerifier-TTS is designed for scenarios involving objects, attributes, and both spatial and non-spatial relations, and it achieves performance levels generally above 0.7. It is important to note that the **examplex in our benchmark are particularly challenging**, featuring highly confounding similar-attribute objects, small objects, occluded objects, and complex compositional spatial relationships (see **Appendix A of the updated manuscript** for detailed examples). Nevertheless, for relatively simpler, general T2I scenarios, OmniVerifier achieves fully acceptable accuracy.
>
> These observations also highlight that even advanced MLLMs currently struggle with highly complex visual-outcome verification. As we scale up model capacity, further improving this capability will be an important direction for future work.

---

> ### Author Response · Authors · 2025-11-20
> **Response to Reviewer Q1VF (Part 2/2)**
>
> >**Q3: While the authors demonstrate in Table 2 that their specifically trained OmniVerifier-7B judge outperforms QwenVL, they do not explore whether a judge with superior foundational abilities would yield further gains. For instance, as shown in Table 1, Gemini-2.5-Pro achieves the highest score on the ViVerBench benchmark. Would a more powerful judge like Gemini-2.5-Pro lead to significantly better TTS results？**
>
> **A3:** Thank you for pointing this out. We have added the results of using Gemini-2.5-Pro as the verifier on T2I-ReasonBench and GenEval++ as follows:
>
> **Table 2: Results of Sequential TTS with Gemini 2.5 Pro on T2I-ReasonBench and GenEval++.**
> | Backbone    | Verifer         | Test-Time Scaling | T2I-ReasonBench Overall | GenEval++ Overall |
> | ----------- | --------------- | ----------------- | ----------------------- | ----------------- |
> | Qwen-Image  | -               | -                 | 55.5                    | 0.675             |
> | Qwen-Image  | Omniverifier-7B | Sequential        | 59.2                    | 0.718             |
> | Qwen-Image  | Gemini 2.5 Pro  | Sequential        | 62.7                    | 0.736             |
> | GPT-Image-1 | -               | -                 | 76.8                    | 0.689             |
> | GPT-Image-1 | Omniverifier-7B | Sequential        | 79.3                    | 0.721             |
> | GPT-Image-1 | Gemini 2.5 Pro  | Sequential        | 81.6                    | 0.746             |
>
>
> Due to Gemini-2.5-Pro’s strong world knowledge and visual-judgment abilities, it achieves significantly better results on both benchmarks. We also find that **incorporating TTS with a strong verifier pushes the baseline into a substantially higher performance tier, markedly extending its achievable upper bound.** Looking ahead, we plan to further improve the model by scaling up both its size and the amount of high-quality training data.
>
> We have added this experiment in **Appendix D of the updated manuscript**. Thank you again for your question.
>
>
> >**Q4: What are the specific failure cases where sequential TTS performs worse than parallel approaches?**
>
> **A4:** This is an excellent question, and we can provide a detailed analysis of the strengths and weaknesses of Sequential TTS versus Parallel TTS.
>
>
>
> Sequential TTS significantly raises the generation performance ceiling of unified model. Through multiple rounds of self-refinement, OmniVerifier continuously leverages its world knowledge, reasoning ability, and critic capability to guide visual generation through fine-grained editing. This allows it to handle highly complex compositional prompts, including domains unseen in the training data. More results are provided in **Fig. 8 of Appendix E in the updated manuscript**. In contrast, for such highly challenging prompts, Parallel TTS struggles to generate accurate outputs regardless of the number of samples. This is because it does not perform targeted reflection and correction for the global prompt condition; its trial-and-error approach without self-reflection makes it difficult to achieve breakthroughs in complex scenarios.
>
>
>
> However, Sequential TTS is **constrained by the backbone’s editing capability and is subject to error accumulation, which affects the final image quality.** **In Fig. 9 of Appendix E**, we present some failure cases of Sequential TTS using GPT-Image-1. While the final image aligns with the prompt, GPT-Image-1 exhibits limited robustness to its own generated image distribution, causing a gradual yellowing of images over successive edits. Although OmniVerifier generally provides correct edit prompts, a weak backbone editing capability prevents accurate application of these changes. Consequently, errors accumulate across multiple rounds of self-refinement, ultimately degrading the final image quality. This highlights that mitigating error accumulation is a critical challenge for future unified models tasked with multi-round generation.

---

### Official Review · Reviewer_Ctie · 2025-10-31

**Soundness:** 3
**Presentation:** 3
**Contribution:** 4
**Rating:** 8
**Confidence:** 4

**Summary:**

This paper introduces the "Generative Universal Verifier," a concept aimed at endowing next-generation multimodal models with the crucial ability to reflect upon and refine their visual outputs. The work presents three core contributions:

1）ViVerBench: A new, comprehensive benchmark spanning 16 task categories designed to evaluate the verification of visual outcomes. Experiments on this benchmark reveal that existing SOTA VLMs significantly underperform human-level capability.

2）OmniVerifier-7B: The authors design two automated data construction pipelines to train a 7B generative verifier, which achieves a substantial +8.3 gain on ViVerBench. Through this process, they identify and analyze three "atomic capabilities" of visual verification: Explicit Alignment, Relational Verification, and Integrative Reasoning.

3. OmniVerifier-TTS: A novel sequential test-time scaling (TTS) paradigm is proposed. This method uses the verifier to bridge image generation and editing within unified models, enabling iterative, fine-grained optimization. This sequential approach is shown to outperform parallel TTS methods like Best-of-N on benchmarks such as T21-ReasonBench and GenEval++.

**Strengths:**

1）Problem Importance: The paper addresses a highly important and timely problem. As MLLMs move towards complex, interleaved reasoning and generation, the ability to self-critique and refine *visual* outputs, not just text, is a fundamental requirement for building more reliable and controllable systems.

2）High-Quality Benchmark: ViVerBench is a major contribution in itself. It is comprehensive, challenging, and meticulously constructed. The evaluations on it provide clear and actionable insights, identifying specific weaknesses in current SOTA models (e.g., mismatched world knowledge, underdeveloped critics for visual reasoning).

3）Deep Capability Analysis: The investigation in Section 4.2 that leads to the identification of three "atomic capabilities" (Explicit Alignment, Relational Verification, Integrative Reasoning) is a standout feature. The discovery that the first two capabilities generalize well, while the third is task-specific, is a deep insight that provides clear guidance for training more efficient and effective universal verifiers.

4）Novel Sequential TTS: The OmniVerifier-TTS framework is a significant methodological advance. It smartly reframes test-time refinement from a parallel *selection* problem (Best-of-N) to a sequential *optimization* problem. By using the verifier to generate concrete "edit prompts," it enables iterative, fine-grained correction, leading to a higher performance ceiling than parallel methods.

**Weaknesses:**

1）Unclear Data Construction Diagram: The diagram illustrating the data construction pipeline in Figure 2 is confusing, particularly for "Method 2: Prompt-fixed Image-Inpainting". As per the flowchart, the line from GPT-5 seems to originate only from the SAM-processed 'True Image'. The 'False Image', generated by FLUX inpainting, lacks a clear connection to the final "Prompt & Explanation" output. This makes the data-pairing process for false examples ambiguous and difficult to follow.

2）Limits of "Universality": The paper honestly acknowledges that the "Integrative Reasoning" capability (e.g., Maze, Robotics) shows minimal cross-task generalization and requires task-specific data. This challenges the "Universal" claim to some extent. The paper would be strengthened by further discussing the implications of this finding for building a *truly* universal verifier.

**Questions:**

1）Addressing Performance Drops: Why does the OmniVerifier-7B model, after being trained with RL on alignment and relational data, show such a performance drop on the 'Chart' task and stagnate on 'Static Physics' (Table 1)? These tasks (especially 'Chart') are prime examples of the atomic capabilities the model was supposedly trained to improve.

2）Rationale for GPT-5 vs. Gemini 2.5 Pro: What was the rationale for using GPT-5 for the visual verification data construction pipelines? Given that your own evaluation (Table 1) identified Gemini 2.5 Pro as the superior model for these verification tasks, were there specific capabilities (e.g., superior prompt generation, better compliance with modification instructions) where GPT-5 was found to be more suitable for the data creation role?

3）Path for Integrative Reasoning: Given the poor generalization of "Integrative Reasoning" tasks, do you see a path toward improving their generalization (e.g., through different training strategies or data augmentation), or do you believe a task-specific "mixture-of-verifiers" model is the more practical future direction?

---

> ### Author Response · Authors · 2025-11-20
> **Response to Reviewer Ctie (Part 1/4)**
>
> We thank you for your thorough review. We appreciate the positive comments, as well as the thoughtful points raised. Please kindly find our response to your comments below. Additionally, all modifications to the manuscript have been highlighted in blue for easy reference. Please feel free to let us know if you have any additional concerns or questions.
>
> >**Q1: Unclear Data Construction Diagram: The diagram illustrating the data construction pipeline in Figure 2 is confusing, particularly for "Method 2: Prompt-fixed Image-Inpainting". As per the flowchart, the line from GPT-5 seems to originate only from the SAM-processed 'True Image'. The 'False Image', generated by FLUX inpainting, lacks a clear connection to the final "Prompt & Explanation" output. This makes the data-pairing process for false examples ambiguous and difficult to follow.**
>
> **A1:** We sincerely apologize for the confusion. Below, we provide a detailed explanation of the workflow of Method 2: Prompt-fixed Image-Inpainting.
>
> **Step 1:** We use SAM 2.1 to segment each complex image, obtaining the mask and bounding box for every object in the image.
>
> **Step 2:** To balance the difficulty distribution of the data, we dynamically select one object based on its mask area.
>
> **Step 3:** We perform inpainting on the true image using the selected mask. Please note that the **inpainting is prompted with “backbone” and uses the object name as a negative prompt.** The purpose is to effectively erase the selected object from the image.
>
> **Step 4:** Using the complex image and the selected bbox as input, we prompt GPT-5 to produce a strict recaption of the true image, with explicit emphasis on the object and attributes inside the bbox. GPT-5 is also asked to **generate an explanation describing what changes in the false (inpainted) image if the object and attributes inside the bbox are removed.**
>
> Steps 3 and 4 are executed **in parallel.** Since inpainting effectively removes the object, the recaption and explanation can be derived solely from the true image and the corresponding bbox.
>
> We provide the full prompts in the **Appendix H of the updated manuscript**, and include all code for the two automated pipelines for visual verifier data construction in `data_construction/method1` and `data_construction/method2` in the **updated supplementary material**. The data can be constructed sequentially according to the `step` indicated in the filenames.

---

> ### Author Response · Authors · 2025-11-20
> **Response to Reviewer Ctie (Part 2/4)**
>
> >**Q2: Limits of "Universality": The paper honestly acknowledges that the "Integrative Reasoning" capability (e.g., Maze, Robotics) shows minimal cross-task generalization and requires task-specific data. This challenges the "Universal" claim to some extent. The paper would be strengthened by further discussing the implications of this finding for building a *truly* universal verifier.**
>
> **A2:** Thank you for your constructive suggestion. Universal Verifier is a specific concept designed for next-generation multimodal systems. We posit that future unified multimodal models require verifiers that are universal and omni-capable. With the scaling of multimodal data, model size, and RL-driven training pipelines, model capabilities will steadily advance, placing them in increasingly challenging and diverse task than those seen today. Unified models will thus require a verifier that can robustly handle diverse reasoning patterns, moving beyond narrowly trained, task-specific modules. This long-term view motivates our effort to design a truly universal verifier.
>
> We identify three fundamental atomic capabilities of a universal visual verifier: *Explicit Alignment*, *Relational Verification*, and *Integrated Reasoning*. Our current work has already made preliminary progress in demonstrating the early generalization properties of a universal verifier with respect to the first two atomic capabilities. However, due to the highly homogeneous domain of the maze training data, we have not yet been able to reveal the full, meaningful generalization of *Integrated Reasoning* capabilities. Nevertheless, we firmly believe that the third atomic capability naturally underlies a broad family of downstream tasks such as Maze, FrozenLake, Snake, and others, all of which rely on the same underlying perception and reasoning mechanism. **This capability is not task-specific; it reflects a stable and transferable mode of cross-modal reasoning.** The current limitations in generalization mainly arise from insufficient data diversity and scale.
>
> Beyond the immediate issues, we also want to discuss how to further advance universality and support broader downstream applications:
>
> - **Integrate more compositional verification datasets** Prior work [1] has shown that atomic capabilities can generalize compositionally. Building on this insight, we plan to construct richer and more challenging datasets to train a universal verifier in broader and more complex settings.
> - **Universal and flexible task interfaces** We do not envision future verifiers as modules that only output global-level explanations or critique. Instead, they must be capable of actively supporting fine-grained judgment for multimodal generation and reasoning. To this end, one of our key goals is the development of multimodal symbolic verifiers that not only output explanations but also support critique with fine-grained symbolic features, such as bounding boxes and points. Moreover, given the emergence of interleaved reasoning and generation, pairwise and sequence-level verifiers will assume growing importance.
>
>
> [1] From f(x) and g(x) to f(g(x)): LLMs Learn New Skills in RL by Composing Old Ones

---

> ### Author Response · Authors · 2025-11-20
> **Response to Reviewer Ctie (Part 3/4)**
>
> >**Q3: Addressing Performance Drops: Why does the OmniVerifier-7B model, after being trained with RL on alignment and relational data, show such a performance drop on the 'Chart' task and stagnate on 'Static Physics' (Table 1)? These tasks (especially 'Chart') are prime examples of the atomic capabilities the model was supposedly trained to improve.**
>
> **A3:** This is an excellent question. Chart task can indeed be viewed as a composition of the first two atomic capabilities. However, the **dataset construction process for charts is particularly challenging**, which limits the performance gains for a 7B-size model. Specifically:
>
> For each matched pair of Python code and image, we ask GPT-5 to perform small but semantically meaningful code modification that must produce visually recognizable changes in the rendered output, rather than merely altering internal data structures. This leads to very difficult samples. Such cross-modal judgment requires not only strong critic capability but also a solid foundation in code understanding and execution, which is especially demanding for a 7B model.
>
> It is also worth noting that in the LaTeX task, we use the same data construction strategy, where GPT-5 introduces extremely subtle modifications to generate falsified code. After training, we observe a **significant improvement on LaTeX (+0.17), which further indicates cross-generalization among atomic capabilities.** Therefore, we attribute the performance drop on Chart to the high difficulty of the data and the limited capability of the 7B backbone. As for the high-difficulty Chart data in ViverBench, we plan to conduct further experiments using a stronger backbone in future work.
>
> Regarding Static Physics, in addition to the first two atomic capabilities, this task crucially relies on world knowledge. Compared with Dynamic Physics, our Static Physics setting includes far more scenarios involving domain-specific knowledge, such as difficult projection problems, advanced electromagnetism questions, etc. These require explicit knowledge supplementation rather than generic visual reasoning. Consequently, OmniVerifier does not observe clear improvements on this task.
>
>
> >**Q4: Rationale for GPT-5 vs. Gemini 2.5 Pro: What was the rationale for using GPT-5 for the visual verification data construction pipelines? Given that your own evaluation (Table 1) identified Gemini 2.5 Pro as the superior model for these verification tasks, were there specific capabilities (e.g., superior prompt generation, better compliance with modification instructions) where GPT-5 was found to be more suitable for the data creation role?**
>
> **A4:** We chose GPT-5 mainly for the following two reasons:
>
> First, our evaluations indicate that GPT-5 provides clear advantages in state awareness, attribute recognition, and grounding, achieving strong accuracy on image captioning tasks.
>
> Second, **GPT-5 offers substantially faster inference than Gemini 2.5 Pro**, which is critical for large-scale data construction .
>
> For these reasons, GPT-5 is adopted as the upstream model in our data construction pipeline.

---

> ### Author Response · Authors · 2025-11-20
> **Response to Reviewer Ctie (Part 4/4)**
>
> >**Q5: Path for Integrative Reasoning: Given the poor generalization of "Integrative Reasoning" tasks, do you see a path toward improving their generalization (e.g., through different training strategies or data augmentation), or do you believe a task-specific "mixture-of-verifiers" model is the more practical future direction?**
>
> **A5:** We argue that addressing the generalization of the third type of atomic capability fundamentally relies on **data augmentation**, including both **diversity-oriented data expansion** and **the construction of compositional datasets**. Prior work [1] has shown that atomic capabilities can generalize through compositional data training. we firmly believe that the third atomic capability naturally underlies a broad family of downstream tasks such as Maze, FrozenLake, Snake, and others, all of which rely on the same underlying perception a n d reasoning mechanism. **This capability is not task-specific; it reflects a stable and transferable mode of cross-modal reasoning.** The current limitations in generalization mainly arise from insufficient data diversity and scale.
>
> In the short term, a task-specific mixture-of-verifiers architecture can indeed achieve strong performance with relatively low implementation cost. However, in the long run, we argue that a **universal verifier** is ultimately necessary for the following reasons:
>
> 1. **Mixture-of-verifiers suffers from inherent Pareto conflicts.**
>
>    Different verifiers (or reward models) naturally impose objectives that may conflict along the Pareto frontier. In complex or highly compositional settings, determining how to weight or select these verifiers becomes increasingly difficult, limiting the scalability and stability of mixture-based approaches.
>
> 2. **Fundamental limits in generalization.**
>
>    As multimodal data, model scale, and RL-driven optimization continue to expand, future multimodal systems will be required to handle tasks far more diverse than those observed today. Designing and maintaining task-specific verifiers for each emerging scenario is unsustainable. A universal verifier with broad generalization is the only viable path forward.
>
> 3. **A true verifier is ultimately a policy model.**
>
>    As shown in **Table 5 of Appendix B in the updated manuscript**, visual-outcome verification already provides measurable gains in generation quality. We posit that verifier models and policy models will eventually converge, enabling a multimodal system capable of *self-evaluation* and ultimately *self-improvement*. A universal verifier is a critical building block toward this direction.

---

### Official Review · Reviewer_YEzL · 2025-11-01

**Soundness:** 3
**Presentation:** 3
**Contribution:** 3
**Rating:** 8
**Confidence:** 3

**Summary:**

The paper targets a missing skill in multimodal LMs: checking whether generated/used images actually match the prompt/reasoning. It builds ViVerBench to show current VLMs lag far behind humans, trains a 7B OmniVerifier via two automatic visual-verification data pipelines, and plugs the verifier into a sequential test-time scaling setup so the model can detect mistakes and issue edit prompts.

**Strengths:**

- Well-motivated problem: visual-outcome verification is clearly under-served and the benchmark exposes real gaps.

- Data pipelines are scalable and produce clean true/false supervision for several atomic verification skills.

- Nice systems angle: using the verifier to improve generation (not just evaluate) via sequential TTS is practical and shows gains.

**Weaknesses:**

- Overstated Universal Claim: The "Universal Verifier" title seems to overstate the model's capabilities, especially given the limitations candidly discussed by the authors. The admission that it "generalize less effectively" to tasks like mazes due to a large domain gap suggests the verifier is still highly dependent on task-specific data distributions.
- Evaluation is mostly on the authors’ benchmark / close tasks; less evidence for robustness on messy, real-world images.

**Questions:**

- In OmniVerifier-TTS, the iterative editing process depends on the “edit prompts.” How are hallucinated or semantically incorrect edit suggestions filtered or corrected during inference?

- For the two automated pipelines you use to construct large-scale visual verification data, how sensitive is the final verifier’s performance to noise or inconsistencies introduced by the upstream models?  and did you consider any robustness or denoising steps beyond the current filtering?

---

> ### Author Response · Authors · 2025-11-20
> **Response to Reviewer YEzL (Part 1/3)**
>
> We thank you for your thorough review. We appreciate the positive comments, as well as the thoughtful points raised. Please kindly find our response to your comments below. Additionally, all modifications to the manuscript have been highlighted in blue for easy reference. Please feel free to let us know if you have any additional concerns or questions.
>
> >**Q1: Overstated Universal Claim: The "Universal Verifier" title seems to overstate the model's capabilities, especially given the limitations candidly discussed by the authors. The admission that it "generalize less effectively" to tasks like mazes due to a large domain gap suggests the verifier is still highly dependent on task-specific data distributions.**
>
> **A1:** Sorry for the misunderstanding. Universal Verifier is a specific concept designed for next-generation multimodal systems. We posit that future unified multimodal models require verifiers that are universal and omni-capable. With the scaling of multimodal data, model size, and RL-driven training pipelines, model capabilities will steadily advance, placing them in increasingly challenging and diverse task than those seen today. Unified models will thus require a verifier that can robustly handle diverse reasoning patterns, moving beyond narrowly trained, task-specific modules. This long-term view motivates our effort to design a truly universal verifier.
>
> We identify three fundamental atomic capabilities of a universal visual verifier: *Explicit Alignment*, *Relational Verification*, and *Integrative Reasoning*. Our current work has already made preliminary progress in demonstrating the early generalization properties of a universal verifier with respect to the first two atomic capabilities. However, due to the highly homogeneous domain of the maze training data, we have not yet been able to reveal the full, meaningful generalization of *Integrative Reasoning* capabilities. Nevertheless, we firmly believe that the third atomic capability naturally underlies a broad family of downstream tasks such as Maze, FrozenLake, Snake, and others, all of which rely on the same underlying perception a n d reasoning mechanism. **This capability is not task-specific; it reflects a stable and transferable mode of cross-modal reasoning.** The current limitations in generalization mainly arise from insufficient data diversity and scale.
>
> Beyond the immediate issues, we also want to discuss how to further advance universality and support broader downstream applications:
>
> - **Integrate more compositional verification datasets.** Prior work [1] has shown that atomic capabilities can generalize compositionally. Building on this insight, we plan to construct richer and more challenging datasets to train a universal verifier in broader and more complex settings.
> - **Universal and flexible task interfaces.** We do not envision future verifiers as modules that only output global-level explanations or critique. Instead, they must be capable of actively supporting fine-grained judgment for multimodal generation and reasoning. To this end, one of our key goals is the development of multimodal symbolic verifiers that not only output explanations but also support critique with fine-grained symbolic features, such as bounding boxes and points. Moreover, given the emergence of interleaved reasoning and generation, pairwise and sequence-level verifiers will assume growing importance.
>
> [1] From f(x) and g(x) to f(g(x)): LLMs Learn New Skills in RL by Composing Old Ones

---

> ### Author Response · Authors · 2025-11-20
> **Response to Reviewer YEzL (Part 2/3)**
>
> >**Q2: Evaluation is mostly on the authors’ benchmark / close tasks; less evidence for robustness on messy, real-world images.**
>
> **A2:** Thank you for your suggestion. We have extended our evaluation to include additional results on other mainstream benchmarks.
>
> For the evaluation of Vision-Language Generative Reward Models, we use VL-RewardBench [2], focusing on the text-outcome verification capability of GenRM.
>
> **Table 1: Evaluation Results on VLRewardBench.**
> | Models  | General  | Hallucination | Reasoning  | Overall Accuracy | Macro Average Accuracy |
> | -- | -- | -- | -- | -- | -- |
> | Qwen 2.5-VL 7B  | 37.16 | 45.79  | 54.40 | 46.72  | 45.79   |
> | OmniVerifier 7B | 41.53(+4.37) | 70.09(+24.3)  | 57.86(+3.46) | 62.80(+16.08)  | 56.49(+10.7) |
>
> We observe that OmniVerifier-7B achieves substantial and comprehensive improvements over the baseline, particularly in reducing hallucination. This shows that **visual outcome verification can effectively generalize to text-outcome verification**, which we attribute to the shared atomic capabilities underlying cross-modal verification.
>
> Furthermore, to examine more general and more complex tasks, we additionally evaluate the models on eight mainstream perception and image-reasoning benchmarks:
>
> **Table 2: Evaluation Results on Mainstream Perception and Image-reasoning Benchmarks.**
> | Models | MMStar  | MMVP  | RealWorldQA | MathVision | EMMA    | VisuLogic  | ZeroBench  | OCRBench   |
> | --- | -- | -- | ---- | --- | -- | --- | --- | --- |
> | Qwen 2.5 VL 7B  | 61.7   | 72.9    | 68.8  | 22.1    | 24.8    | 26.9  | 13.7 | 85.1   |
> | OmniVerifier 7B | 63.9(+2.2) | 77.7(+4.8) | 68.1(-0.7)  | 25.2(+3.1) | 29.4(+4.6) | 25.4(-1.5) | 14.4(+0.7) | 87.1(+2.0) |
>
>
> OmniVerifier demonstrates clear improvements on most benchmarks. This suggests that **a critic/verifier model trained with RL on pointwise samples can generalize into a strong policy model, leading to broad gains across diverse downstream generation tasks.** This finding aligns with observations from LLaVA-Critic-R1 [3], and we further validate it in the visual-outcome setting.
>
> We argue that this observation, namely critics enhance generation, highlights a highly promising direction for future large-scale model development: **a strong policy model can be trained into a strong verifier or critic model, and the two can mutually reinforce one another, enabling self-improvement and self-evolution.** This also highlights the importance of a universal verifier.
>
> We have added these experiments in **Appendix B of the updated manuscript**. Thank you again for your valuable feedback.
>
> >**Q3: In OmniVerifier-TTS, the iterative editing process depends on the “edit prompts.” How are hallucinated or semantically incorrect edit suggestions filtered or corrected during inference?**
>
> **A3:** This is an excellent question. Before addressing it directly, we would like to share an insight from our training analysis. Although our RL setup applies rule-based rewards only to the binary true/false answers without any explicit supervision on explanations or edit prompts, we were surprised to observe that the **model’s explanation quality, including edit-prompt generation, improved and generalized significantly**. As shown in **Appendix F of the manuscript**, OmniVerifier also naturally learns a high-quality and well-structured LongCoT pattern, to decompose a complex problem into a series of specific and simpler subproblems.
>
>
> In OmniVerifier-TTS, hallucinated or semantically incorrect edit prompts are naturally filtered through the design of our verification–editing loop. Our method does not rely on any external model; instead, the verifier itself serves as the filtering mechanism. Concretely:
>
> - **Edit prompts are automatically validated by OmniVerifier in the next iteration.** Each round produces a new image, which is evaluated by OmniVerifier using its binary True/False decision, explicit alignment analysis, and structured explanation. If an edit prompt is incorrect or hallucinatory, the resulting image will still be classified as False in the next iteration, prompting the verifier to generate a corrected edit suggestion. In this way, **incorrect edit prompts cannot propagate, they are exposed and corrected in the next iteration, forming a self-correcting, closed-loop refinement cycle that compensates for errors.**
>
> - **Convergence is governed by the True/False decision.** Iterations terminate only when the verifier itself returns a True judgement. This produces a closed-loop correction mechanism rather than relying on trust in any single edit suggestion.
>
> - **Edit prompts can follow a minimal-edit principle.** In principle, the verifier can be guided to propose small, localized, and atomic corrections, which are easier to validate and less susceptible to semantic drift.
>
>
>
> [2] VL-RewardBench: A Challenging Benchmark for Vision-Language Generative Reward Models
>
> [3] LLaVA-Critic-R1: Your Critic Model is Secretly a Strong Policy Model

---

> > ### Comment · Reviewer_YEzL · 2025-11-27
> >
> > Thank you for your response; this has resolved my concerns.

---

> ### Author Response · Authors · 2025-11-20
> **Response to Reviewer YEzL (Part 3/3)**
>
> >**Q4: For the two automated pipelines you use to construct large-scale visual verification data, how sensitive is the final verifier’s performance to noise or inconsistencies introduced by the upstream models? and did you consider any robustness or denoising steps beyond the current filtering?**
>
> **A4:** In principle, noise from upstream models (GPT-5 recaptioning, SAM segmentation, FLUX inpainting) could introduce inconsistencies. However, our empirical observations indicate that the final verifier is only **weakly sensitive to such noise.**
>
> Our constructed data is inherently contrastive, with a clear separation between True and False pairs, making the supervision direction robust to upstream noise. In addition, the Seed1.5-VL Best-of-10 filtering (threshold ≥ 0.6) effectively removes incorrect samples.  Empirically,  although the RL objective supervises only the true/false label, the model’s explanations and edit prompts become more stable and precise (Appendix F), which also indicates that noise in the synthetic data does not degrade performance, but instead reinforces alignment-oriented behavior. The model also exhibits atomic-capability generalization, transferring from training on tasks *Object* and *Attribute* to unseen tasks such as *Spatial* and *LaTeX*. Such cross-domain generalization would be unlikely if the training data were dominated by noise.
>
>
> We did consider additional denoising strategies beyond the current Seed1.5-VL Best-of-10 filtering. In particular, we prioritize (1) **Cross-Model Verification** and (2) **Atomic-question Consistency Filter**.
>
> For Cross-Model Verification, we employ multiple heterogeneous, high-capability models (e.g., GPT-5, Gemini 2.5 Pro, Grok 4.1) as judges and retain only the samples on which all models agree. This effectively removes biases or incorrect from any single model.
>
> For Atomic-question Consistency Filter, each prompt is first decomposed into a set of atomic visual questions (object existence, color, count, spatial relations, etc.). A strong MLLM (e.g., GPT-5) is then asked to answer these atomic questions based solely on the image; a sample is kept only when the predicted answers match the expected values. This method is highly effective at detecting upstream noise—such as incorrect recaptions, inpainting artifacts, or semantic drift—and naturally aligns with our atomic-capability formulation.

---

### Author Response · Authors · 2025-11-20
**Global Response**

We sincerely thank all the reviewers for their thorough reviews and valuable feedback. We are glad to hear that we addresses an important and timely problem (Reviewer YEzL, Ctie), high-quality benchmark and data construction: (all reviewers), the experimental analysis is thorough and presents a novel approach (all reviewers).

We summarize our responses to the reviewers' comments as follows:

- We additionally provide more evaluation on mainstream benchmarks to show the improvement of OmniVerifier and **updated our manuscript in the Appendix. B** (Reviewer YEzL)
- We additionally conduct an in-depth analysis of OmniVerifier-TTS, including its runtime efficiency and a detailed examination of its strengths and weaknesses, and **updated our manuscript in the Appendix. C, D, and E** (Reviewer Q1VF)
- We provide additional insights and analysis regarding the potential capabilities of future universal verifiers and their possible applications. (Reviewer YEzL, and Ctie)

In addition, we would like to take this opportunity to discuss and summarize what future Universal Verifiers could achieve and the potential benefits they may bring:

1. **Verifier as a strong policy model.** In the future, we envision that the verifier will no longer be a standalone model. Instead, we aim to train powerful judgment capabilities on top of strong policy models, with the two complementing each other. We expect a multimodal judge that not only evaluates text outcomes but also demonstrates strong generalization across visual and omni-modality outcomes. We point out that self-improvement drives model progress, enabling self-evaluation and self-refinement without relying on labels, ultimately achieving self-evolution. OmniVerifier represents the first step toward this vision.
2. **For traditional Text-to-Image generation.** OmniVerifier has the potential to enable fully automated end-to-end evaluation. Automated evaluation of text-to-image generation has historically been highly inaccurate, with precise assessment heavily dependent on human effort, which greatly limits further progress in T2I generation. We have been exploring how advanced vision-language models can facilitate full-process evaluation, and in the future, we plan to invest more in model scaling and exploring stronger training paradigms to advance toward this goal.
3. **For the future era of interleaved reasoning and generation.** Crucially, a full-modality hybrid judge is essential, moving beyond the constraints of text-outcome verification. To address the current limitations in assessing visual outcomes, we are steering OmniVerifier toward sequence-level cross-modal verification. This development is particularly vital for complex tasks such as game reasoning, GUI interaction, and visual narratives.

We reply to each reviewer's questions in detail below their reviews. Please kindly check them out. Thank you and please feel free to ask any further questions.

---

### Meta-Review · Area_Chair_CvWW · 2026-01-06

**Summary:**

This submission received three scores of 8, indicating a clear, consistent, and strong acceptance recommendation. Reviewers agreed that the paper addresses an important problem in multimodal reasoning and makes strong contributions through a new benchmark, scalable data construction pipelines, and a novel sequential test-time refinement framework.

The Area Chair concurs with this assessment and supports oral acceptance.

**Reviewer Concerns:**

The rebuttal successfully addressed the majority of reviewer questions and clarifications, including evaluation breadth, data construction details, robustness to noise, and the practical behavior of the sequential test-time scaling framework.

The only outstanding concerns is the computational cost considerations. The authors added a detailed runtime and efficiency analysis for OmniVerifier in rebuttal, which clarifies the cost-performance tradeoff and indicates computational overhead remains a non-trivial challenge. This can be discussed in the future work section.

**Reviewer Scores:**

Based on the discussion and reviewer follow-up comments, the Area Chair estimates that all three reviewers would maintain their original scores of 8 if they had participated fully in the post-rebuttal discussion.

Original Average Rating: 8.00 (Min: 8, Max: 8)

Estimated Final Average Rating: 8.00 (Min: 8, Max: 8)

---

### Decision · Program_Chairs · 2026-01-26

Accept (Oral)